# Aging drives cerebrovascular network remodeling and functional changes in the mouse brain

Hannah C. Bennett[1,7], Qingguang Zhang ®[2,3,7], Yuan-ting Wu[1,5,7], Steffy B. Manjila[1,7], Uree Chon[1,6], Donghui Shin ®[1], Daniel J. Vanselow[1], Hyun-Jae Pi[1], Patrick J. Drew ®[2,4] & Yongsoo Kim ®[1,2] ✉

Aging is frequently associated with compromised cerebrovasculature and pericytes. However, we do not know how normal aging differentially impacts vascular structure and function in different brain areas. Here we utilize mesoscale microscopy methods and in vivo imaging to determine detailed changes in aged murine cerebrovascular networks. Whole-brain vascular tracing shows an overall ~10% decrease in vascular length and branching density with ~7% increase in vascular radii in aged brains. Light sheet imaging with 3D immunolabeling reveals increased arteriole tortuosity of aged brains. Notably, vasculature and pericyte densities show selective and significant reductions in the deep cortical layers, hippocampal network, and basal forebrain areas. We find increased blood extravasation, implying compromised blood-brain barrier function in aged brains. Moreover, in vivo imaging in awake mice demonstrates reduced baseline and on-demand blood oxygenation despite relatively intact neurovascular coupling. Collectively, we uncover regional vulnerabilities of cerebrovascular network and physiological changes that can mediate cognitive decline in normal aging.

Aging is the primary risk factor for the development of various neurodegenerative diseases[1]. Notably, aging is associated with decreased cerebral blood flow and general vascular impairment. A common denominator in diseases that increases the risk of dementia, such as stroke, atherosclerosis, and diabetes mellitus, is vascular perturbation and dysfunction of neurovascular coupling[2–9]. All of the disease processes mentioned above increase the risk of developing vascular dementia, which is the second leading cause of cognitive impairment in the United States. Impairment in the cerebrovascular network can have a significant impact on energy supply and metabolic waste removal processes, which can result in neuronal death linked with

various clinical symptoms[10–12]. Thus, understanding the anatomical and functional changes in the brain vasculature upon normal aging is a critical first step in understanding neurodegenerative disorders.

The vessels of the cerebrovascular network are composed of endothelial cells linked by tight junctions. These blood vessels are surrounded by mural cells, such as vascular smooth muscle cells and pericytes, which wrap around vessels of the vascular tree and contribute to blood flow regulation[13]. Pericytes are essential for maintaining the blood-brain barrier and play important roles in waste removal and capillary blood flow regulation[14–16]. The importance of these vascular cell types is becoming increasingly recognized in the

[1]Department of Neural and Behavioral Sciences, The Pennsylvania State University, Hershey, PA 17033, USA. [2]Center for Neural Engineering, Department of Engineering Science and Mechanics, The Pennsylvania State University, University Park, PA 16802, USA. [3]Department of Physiology, Michigan State University, East Lansing, MI 48824, USA. [4]Department of Biomedical Engineering, Biology, and Neurosurgery, The Pennsylvania State University, University Park, PA 16802, USA. [5]Present address: Department of Neurosurgery, Department of Computational Biomedicine, Cedars-Sinai Medical Center, Los Angeles, CA 90048, USA. [6]Present address: Neurosciences Graduate Program, Stanford University, Stanford, CA 94305, USA. [7]These authors contributed equally: Hannah C. Bennett, Qingguang Zhang, Yuan-ting Wu, Steffy B. Manjila. ✉e-mail: yuk17@psu.edu

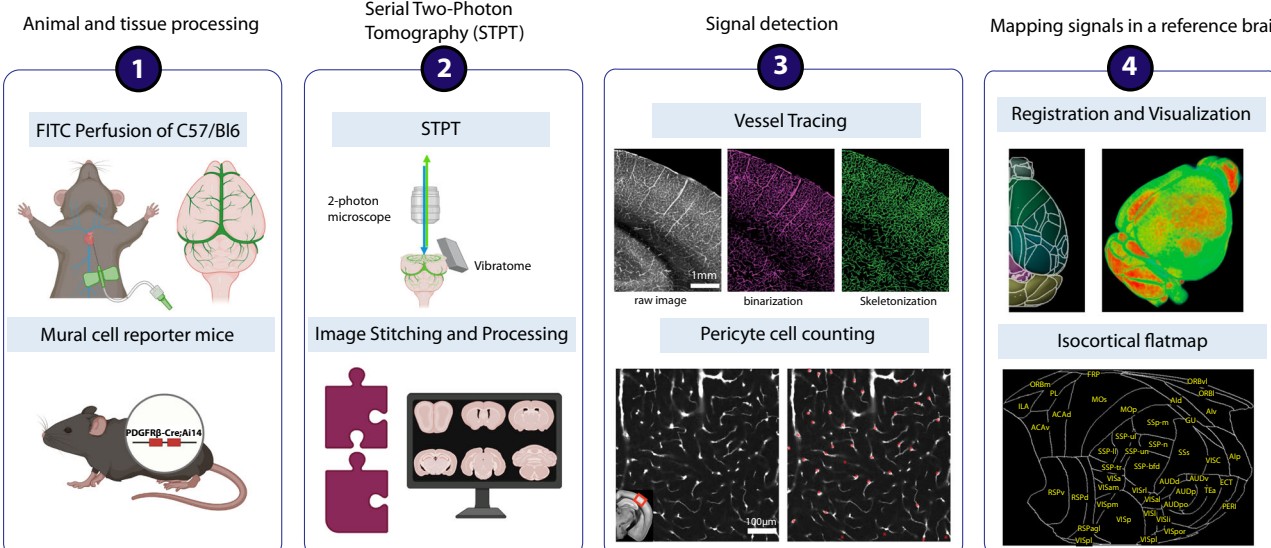

**Fig. 1 | STPT based vessel tracing and pericyte counting pipeline.** The steps of the STPT pipeline are outlined in order. **1**. Brain sample collection: vascular filling procedure via cardiac perfusion with FITC conjugated albumin gel and pericyte mapping using PDGFRβ-Cre; Ai14 reporter mice. **2**. STPT imaging: a combination of 2-photon microscope 2D tile imaging with serial vibratome sectioning to obtain cellular resolution images of the whole mouse brain. These image tiles are then stitched to reconstruct tissue sections spanning the olfactory bulb to the cerebellum. **3**. Signal detection: vascular tracing with binarization of FITC-filled vascular signals and skeletonization and deep learning-based detection of capillary pericytes. **4**. Mapping signals in a reference brain: All detected signals were registered to the Allen Common Coordinate Framework (Allen CCF) and an isocortical flatmap was used to examine signals in the isocortex.

context of brain disorders, particularly in the case of neurodegenerative diseases. Previous studies have shown that aging with cognitive impairment is associated with vascular pathologies, including increased arterial tortuosity, rarefaction of the vascular tree, and impairment of pericyte dynamics[3,5,14,17–19]. In addition to anatomical changes, advanced aging is associated with reduced cerebral blood flow (CBF), increased CBF pulsatility, and stiffening of the major arteries[20–22]. It is becoming increasingly recognized that disruption to the brain's vasculature may precede the neuronal damage associated with neurodegenerative disease and other types of dementia[23], implying that vascular dysfunction may play a causative role in neurodegeneration. Despite its significance, it remains unclear how the cerebrovascular network and mural cell types across different brain regions undergo structural and functional changes during the aging process. Prior work has primarily focused on single brain regions without accounting for brain-wide changes in the cerebrovascular network, largely due to the complexities of visualizing and analyzing large 3D brain volumes.

Recent advances in 3D whole brain imaging methods make it possible to quantitatively examine detailed cerebrovascular networks in the entire mouse brain[24–29]. We previously showed that regional differences in pericyte density and cerebrovascular structure strongly correlate with the number of parvalbumin-expressing neuron populations in the cortex of young adult (2-month-old) mice[29]. Here, leveraging high-resolution 3D mapping methods (light sheet and serial two-photon microscopy), we ask whether there are regional vulnerabilities within the cerebrovasculature and mural cell types upon aging at 18 months (early aging), and 24-month-old (late aging) following the JAX Lifespan as biomarker criteria[30]. 24 months of age was deemed as late aging without a significant death rate, as mouse survivorship steadily declines after 24 months old[30]. We found selective reduction of vascular length and pericyte density in deep cortical layers, as well as the basal forebrain areas where cholinergic neurons with large cell bodies reside. Aging also causes vascular remodeling with increased arterial tortuosity in the isocortex and reduces capillary pericyte density in the entorhinal cortex. In addition to anatomical changes, in vivo imaging (two-photon and wide field optical intrinsic signal

imaging) of the vasculature in awake aged mice indicates low blood oxygenation at baseline and evoked conditions. Collectively, our results demonstrate significant cerebrovascular network changes, linked to regional vulnerabilities and reduced hemodynamic responsiveness in aging.

## Results
### Early aging in the mouse brain shows overall decreased vascular length density and branching density, but increased vascular radii

To determine structural changes of the cerebrovasculature upon normal aging, we applied our cerebrovascular mapping pipelines in 18-month-old (aged) mice in order to compare 2-month-old (young adult) mice[29] (Fig. 1). We labeled the brain vasculature by cardiac perfusion of fluorescein isothiocyanate (FITC)-conjugated albumin gel[27,29,31,32]. Then, we utilized serial two-photon tomography (STPT) imaging to image the whole mouse brain at 1x1x5 µm resolution (x,y,z; media-lateral, dorsal-ventral, rostral-caudal) followed by computational analysis for vasculature tracing and quantification[29,33]. All signals were registered to the Allen Common Coordinate Framework (AllenCCF) as a reference brain[34] (Fig. 1).

To identify potential regional vascular vulnerabilities, we first examined the overall changes of the cerebrovasculature across the whole mouse brain, comparing 18-month-old mice to 2-month-old mice (Fig. 2A). Total vessel length in most regions remained similar between 18- and 2-month-old mice (Fig. 2B) but overall brain volume increased about 6% (Fig. 2C; not statistically significant). The brain volume increase was also seen with in vivo longitudinal MRI[35], indicating our result is not an artifact from fixation or imaging. As a result, overall vascular length density across different brain regions decreased by 5 – 10% in the aged brain (Fig. 2D). In addition, we found an approximate 10-20% decrease in branching density across most brain regions (Fig. 2E; Source Data File). In contrast, the average radius of 18-month-old mouse brain vasculature is increased by about 5 – 10% compared to 2-month-old mice, suggesting reduced basal constrictive tone (Fig. 2F; Source Data File). Notably, we found significant changes in brain regions related to memory processing and storage (e.g.,

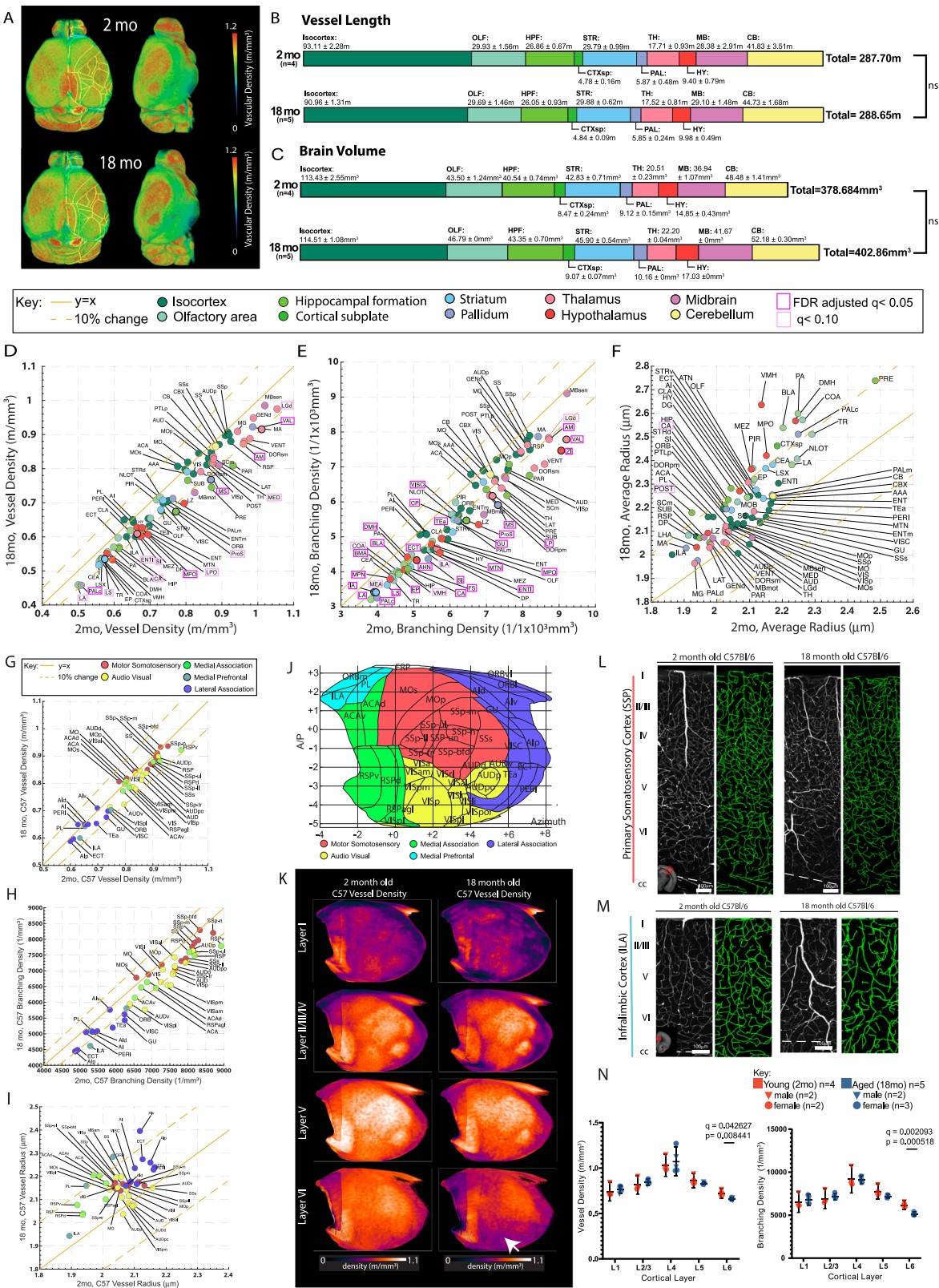

Ammon's horn; CA, lateral entorhinal cortex; ENTl, Anteromedial nucleus; AM), appetitive behavior (e.g., medial preoptic area; MPO, ventral premammillary area; PMv), body physiology and sleep (e.g., lateral preoptic area; LPO, anterior hypothalamic area; AHA), attention (e.g., substantia innominata; SI, medial septum; MS as basal forebrain areas), sensory processing and integration (e.g., zona incerta; ZI, Dorsal lateral geniculate nucleus; LGd), and executive function (e.g.,

medial group of the dorsal thalamus; MED) (Fig. 2D–F, highlighted with magenta boxes; Source Data File).

Next, we examined isocortical areas for aging-related vascular changes. Surprisingly, isocortical areas showed no significant changes, with mostly less than 10% decrease in length and branching density, and about 5% increase in average vessel radius (Fig. 2G–I). To examine vascular changes in the isocortex more intuitively, we utilized our

**Fig. 2 | Region specific reduction of vascular length and branching density and increased vascular radii. A** The averaged vasculature length density of 2-month-old (2mo, $N = 4$) and 18-month-old (18mo, $n = 5$) brains registered to the Allen CCF. **B, C** Summed vessel length (**B**) and brain volume (**C**) in 2-month-old and 18-month-old brains with the mean and SD provided for each anatomical region. **D–F** Scatter plots of averaged vascular length density (m/mm³) (**D**), vascular branching density (1/mm³) (**F**), and vascular radii (μm) between 2-month-old (x axis) and 18-month-old (y axis) brains across different brain regions. Areas that reach FDR adjusted $q < 0.05$ are outlined in magenta boxes. **G–I** Scatter plots of the isocortex data for averaged vascular length density (**G**), vascular branching density (**H**), vascular radii (**I**) between 2-month-old (x axis) and 18-month-old (y axis) brains. Isocortical areas are color coded based on grouping in (**J**). The solid yellow line represents y = x and the dotted line on either side represents a 10% difference from the solid yellow line. No areas show significant changes. **J** Isocortical flatmap with Allen CCF border lines and region-based color-coding. Y axis: Bregma anterior-posterior (**A–P**) coordinates, X axis: the azimuth coordinate represents the physical distance by tracing the cortical surface on the coronal cut. **K** Averaged vascular length density of different cortical layers in the flatmap between 2-month-old and 18-month-old brains. A white arrow highlights the significant decrease of vascular length density in the layer 6 cortical layers. **L, M** Representative example 250μm maximum intensity projection images of the primary somatosensory cortex (**L**) and the infralimbic cortex (**M**) with vascular tracing (green on the right side) between 2-month-old ($n = 4$) and 18-month-old brains ($n = 5$). Note the significant reduction of vasculature in the deep layer. **N** Both vascular length and branching density showed reductions in layer 6. Scale bars for **L** and **M** are 100 μm. For two group comparisons, two-sided unpaired t-tests (p-value) were used with multiple comparison correction (q values) to account for comparisons across multiple brain regions (**B–I, N**). Data are presented as mean values +/- SD for all graphs. Source data are provided as a Source Data file. Brain region abbreviations can be found in the Source Data File or Allen atlas at https://atlas.brain-map.org/atlas?atlas=602630314.

previously developed isocortical flatmap with five distinct cortical domains marked by different colors (Fig. 2J)[29]. We found a significant reduction in vessel length density only in layer 6 of aged brains compared to young brains (Fig. 2K–N). Moreover, we quantified the nearest neighbor distance to vessels as a metric to access blood supply and found a significant increase only in layer 6 in 18-month-old mouse brain (Supplementary Fig. 1). Our result corroborates a previous finding showing selective vulnerability of deep cortical layers to aging[36,37]. Together, these findings indicate that the vasculature of the isocortex is relatively resilient to aging, and the earliest evidence of age-related vascular degeneration occurs in layer 6.

## Pericyte density in aged brains showed significant decrease in basal forebrain regions and the deep cortical layer

Pericytes are a mural cell type that plays a key role in the regulation of the capillary network blood flow and diameter and are known to be vulnerable in aging[15,16,38,39]. Our results show increased vascular radius in aged brains, which raises the possibility of dysfunction in pericytes in the maintenance of vascular diameter. To quantitatively determine changes of pericytes, we compared capillary pericyte densities in 2-month-old and 18-month-old PDGFRβ-Cre;Ai14 mice[40,41], where tdTomato is expressed in pericytes and other mural cells. We used STPT imaging of PDGFRβ-Cre;Ai14 mice with previously developed computational analyzes to image, identify, and quantify changes of capillary pericytes upon aging across the whole mouse brain[29,33] (Fig. 1 bottom and Fig. 3A).

Overall, pericyte density in the aged brain remained within 10% of that in young brains in most areas, including many cortical and thalamic subregions (Fig. 3B; Source Data File). However, a significant reduction of pericyte density was found in basal forebrain areas (e.g., the substantia innominata; SI, magnocellular nucleus; MA) and the closely related anterior amygdala area (AAA) (Fig. 3B; red boxed, C-D)[42]. Considering the basal forebrain contains cortical-projecting cholinergic neurons, the observed significant reduction in pericyte and vascular densities reflects the selective and early vulnerability of the basal forebrain during aging. These results could potentially provide a link between known vascular impairment and dysfunction of cholinergic neurons in neurodegenerative diseases such as Alzheimer's disease[43,44].

Given that we saw few vascular changes with aging in the isocortex (Fig. 2), we investigated whether this resilience extends to pericyte density. We compared 2-month- and 18-month-old mice capillary pericyte densities by brain region using our isocortical flatmap (Fig. 3E). The capillary pericyte density in aged mice overall remained similar to, or even slightly increased as compared with young adult mice (Fig. 3E, F), particularly in motor sensory regions (white and gray arrowheads in Fig. 3E). Due to reduced vessel length density, the overall pericyte cell body coverage (capillary pericyte number per vascular length) is increased by about 10% in sensorimotor areas in

aged mice compared to young adult mice (Fig. 3G). In contrast to sensorimotor areas, relatively little or even reduced pericyte density and coverage was observed in medial prefrontal areas (Fig. 3E white arrow, 3F-G), suggesting regionally distinct vulnerabilities of pericytes with aging.

We then asked whether there are selective changes across cortical layers. We noted that the deep cortical layer (L6) showed a selective reduction of pericyte density in the infralimbic cortex, while the superficial layers (2/3 and 4) in the whisker representation of the primary somatosensory cortex ('barrel field') showed a significant increase in the 18-month-old brain compared to the 2-month-old brain (Fig. 3H–J). When layer-specific density from all isocortical areas was combined, pericyte density was significantly reduced in deep layer 6b in the aged brain, while layers 2/3 and 4 showed significant increases (Fig. 3K). Considering layer 6b plays a role in brain state modulation[45] and the protective role of pericytes in vascular integrity, significant reduction of the pericytes could make layer 6b and nearby white matter tracks more vulnerable upon aging[37,46].

## Artery specific labeling shows striking vascular remodeling in penetrating cortical arterioles of aged brains

Previous studies have identified age-related changes in arteries and arterioles in both rodents and humans[47,48]. To investigate potential remodeling in main arteries and penetrating cortical arterioles, we utilized tissue clearing, 3D immunolabeling, and high-resolution light sheet fluorescence microscopy (LSFM) imaging (Fig. 4A) (see Methods for more details). We labeled arteries with smooth muscle actin (Acta 2) and transgelin (Sm22) antibodies, pan-vasculature with lectin, and pericytes with CD13 and PDGFRβ antibodies in the same brain. This approach enabled us to examine different vascular compartments and mural cell types in the same intact 3D brain (Fig. 4B–H). Despite the volume shrinkage due to dehydration-based tissue clearing methods, we confirmed that the overall vascular geometry was maintained by comparing in vivo two-photon and LSFM imaging from the same animal (Supplementary Fig. 2).

We applied the method to 2-month-old and 24-month-old (late aging) C57BL/6 mice to test whether late aging shows structural remodeling of different vascular compartments and progression of capillary pericyte density reduction. We first focused our analysis on the middle cerebral artery and anterior communicating artery branches contributing to the anterior circulation of the circle of Willis, which is responsible for supplying the majority of cerebral blood flow (Fig. 5A). We quantified the average radius of each artery. We did not find significant differences in young and aged mice, nor differences between sexes (Fig. 5B, C), suggesting that aging does not impact the diameters of the main arteries of the anterior brain circulation.

Next, we examined the number of cortical penetrating arterioles, which are bottlenecks in the supply of blood to the brain[49,50]. There were no significant changes in cortical arteriole numbers

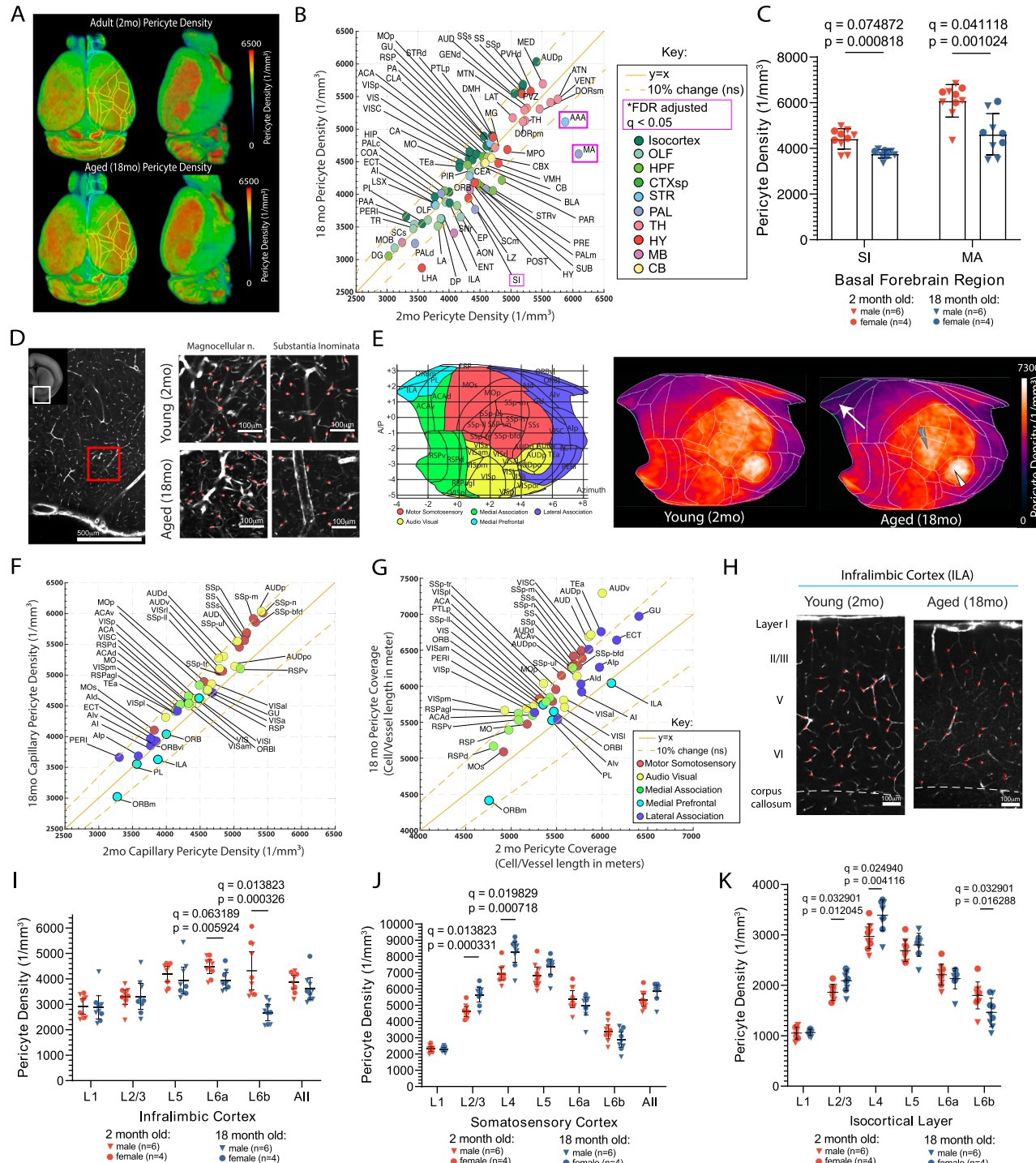

**Fig. 3 | Aged brain showed selective reduction of capillary pericytes in the basal forebrain area and the deep cortical layer. A** Averaged capillary pericyte density in 2-month-old ($n = 10$) and 18-month-old ($n = 9$) PDGFRβ-Cre;Ai14 mouse brains that are registered to the Allen CCF. **B** A scatter plot of capillary pericyte density between 2-month-old (x axis) and 18-month-old (y axis). Brain areas are color coded based on the Allen CCF ontology. Brain regions that are significantly different between groups, reflected by FDR adjusted $q < 0.05$, are outlined in magenta boxes. The solid yellow line represents the value for y = x and the yellow dotted lines on either side of the solid yellow line represent a 10% difference from the solid yellow line. **C** Bar graphs of capillary pericyte density in the substantia innominata and magnocellular nucleus between 2-month-old and 18-month-old brains. **D** Representative images of the basal forebrain, scale bar 500 μm(left) and higher resolution examples of the magnocellular nucleus and substantia innominata in

2-month-old and 18-month-old brains, scale bar 100 μm. Red dots represent detected pericyte cell bodies in each respective region. **E** The isocortical flatmap (left) and averaged capillary pericyte densities plotted in the flatmap from 2-month-old and 18-month-old brains. **F**, **G** Scatter plots of capillary pericyte density (**F**) and pericyte coverage (pericyte density per vascular length density; **G**) in isocortical areas. **H** Representative images of capillary pericyte density in the infralimbic cortex from 2-month-old and 18-month-old brains, scale bar 100 μm. **I–K** Layer specific capillary pericyte densities from the infralimbic cortex (**I**), the somatosensory cortex (**J**), and across all isocortical areas (**K**). Note the significant density reduction in the deep cortical layers. For two group comparisons, two-sided unpaired t-tests (p-value) were used with multiple comparison correction (q values) to account for comparisons across multiple brain regions (**B**, **C**, **F**, **G**, **I–K**). Data are presented as mean values +/- SD for all graphs. Source data are provided as a Source Data file.

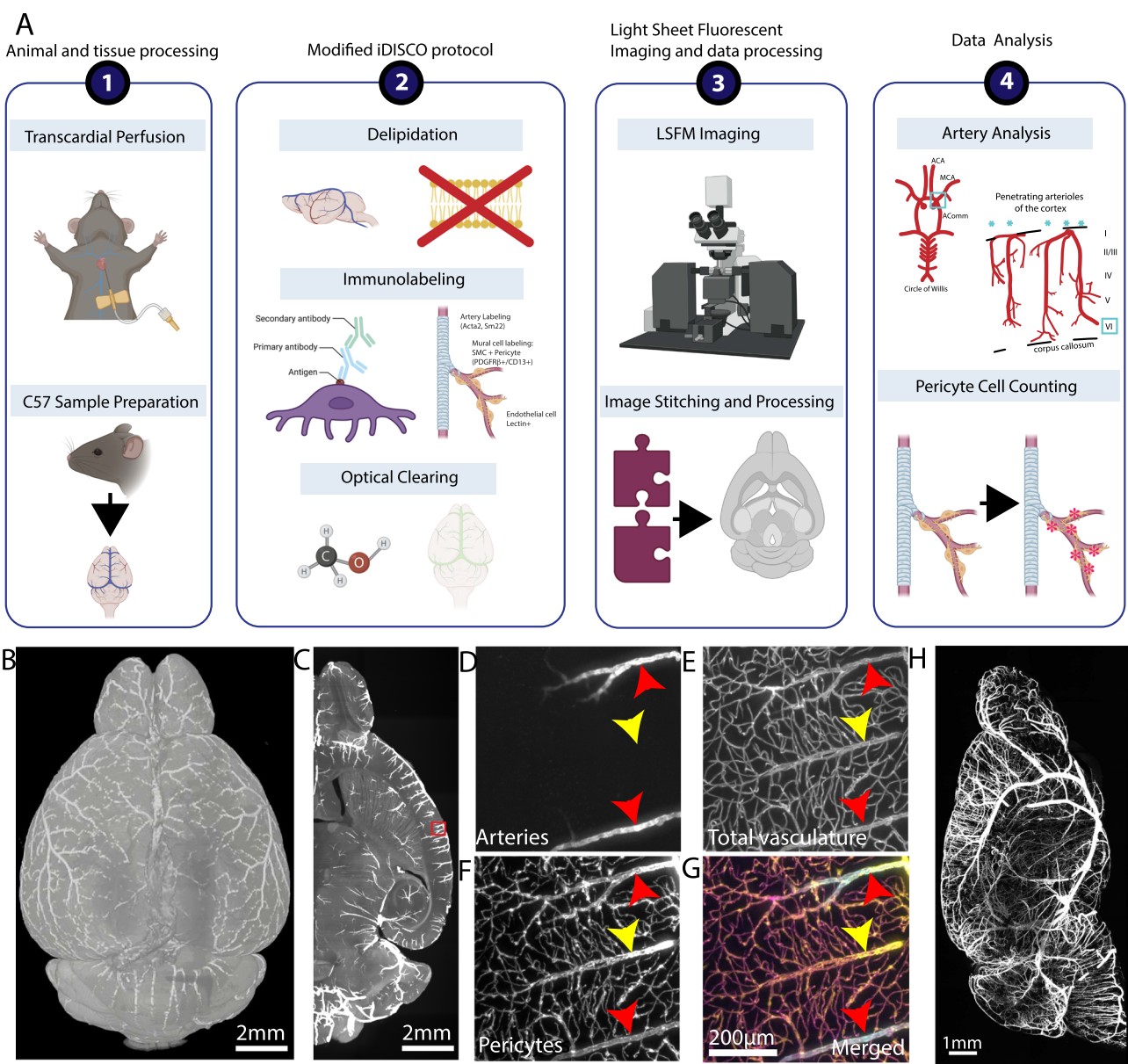

**Fig. 4 | Tissue clearing and 3D immunolabeling with LSFM imaging to examine different vascular compartments and mural cells in the same brain. A** The steps of brain clearing, whole brain immunolabeling, and light sheet fluorescent microscopy (LSFM) pipeline are outlined in order. **1**. Brain sample collection with transcardial perfusion. **2**. Modified iDISCO protocol including delipidation, immunolabeling for arteries, whole vasculature, and pericytes, and optical clearing. **3**. LSFM imaging and data processing to visualize cleared brains at cellular resolution. **4**. Data analysis such as arteriole geometry analysis and pericyte counting.

**B** 3D reconstruction of a brain with artery staining by LSFM imaging, scale bar 2 mm. **C** Max projection of the 500 μm thick z stack of the artery staining, scale bar 2 mm. **D**−**G** Zoom-in images of the red box area from (**C**), scale bars 200 μm. **D** artery staining in the green channel, **E** lectin based total vasculature staining in the red channel, **F** pericyte staining in the far-red channel, **G** a merged image of pseudo-colored arteries (blue), total vasculature (green), and pericyte (red). **H** Maximum projection of the artery channel in a brain hemisphere, scale bar 1 mm.

(both total arterioles and arterioles that extend into layer 6/corpus callosum) in aged brains compared to the young adult mice (Fig. 5D). However, we observed highly tortuous (twisted) vessels across the entire cortex (Fig. 5E; highlighted with red arrowheads), which is consistent with prior observations in aged animals and humans[51,52]. Further analysis revealed that aged animals demonstrate increased arteriole tortuosity, as measured by the arc chord ratio (Fig. 5F) (see Methods for more details). The number of branching points per arteriole remains similar across the age group (Fig. 5G). This increased tortuosity of penetrating arterioles will result in increased blood flow resistance, leading to slowed blood flow with decreased oxygen and nutrient delivery if there is no increase in blood pressure. This decreased flow could make the deep cortical layers and nearby

white matter tracks vulnerable during aging[37]. Lastly, we performed vascular tracing using pan-vascular lectin labeling and found no significant difference in vascular length density in the isocortex except a significant reduction in the infralimbic cortex in the 24-month-old mice (Supplementary Fig. 3). Moreover, we found that only the cortical layer 6 shows a significant reduction in the vascular length density in the 24-month-old mice (Supplementary Fig. 3), similar to changes identified in the 18-month-old mice (Fig. 2N).

**Advanced aging is associated with selective loss of capillary pericytes and blood brain barrier impairment**
Different pericyte subtypes associate with different vascular branches (Fig. 5H)[13]. Aging has been shown to impair specific pericyte subtypes,

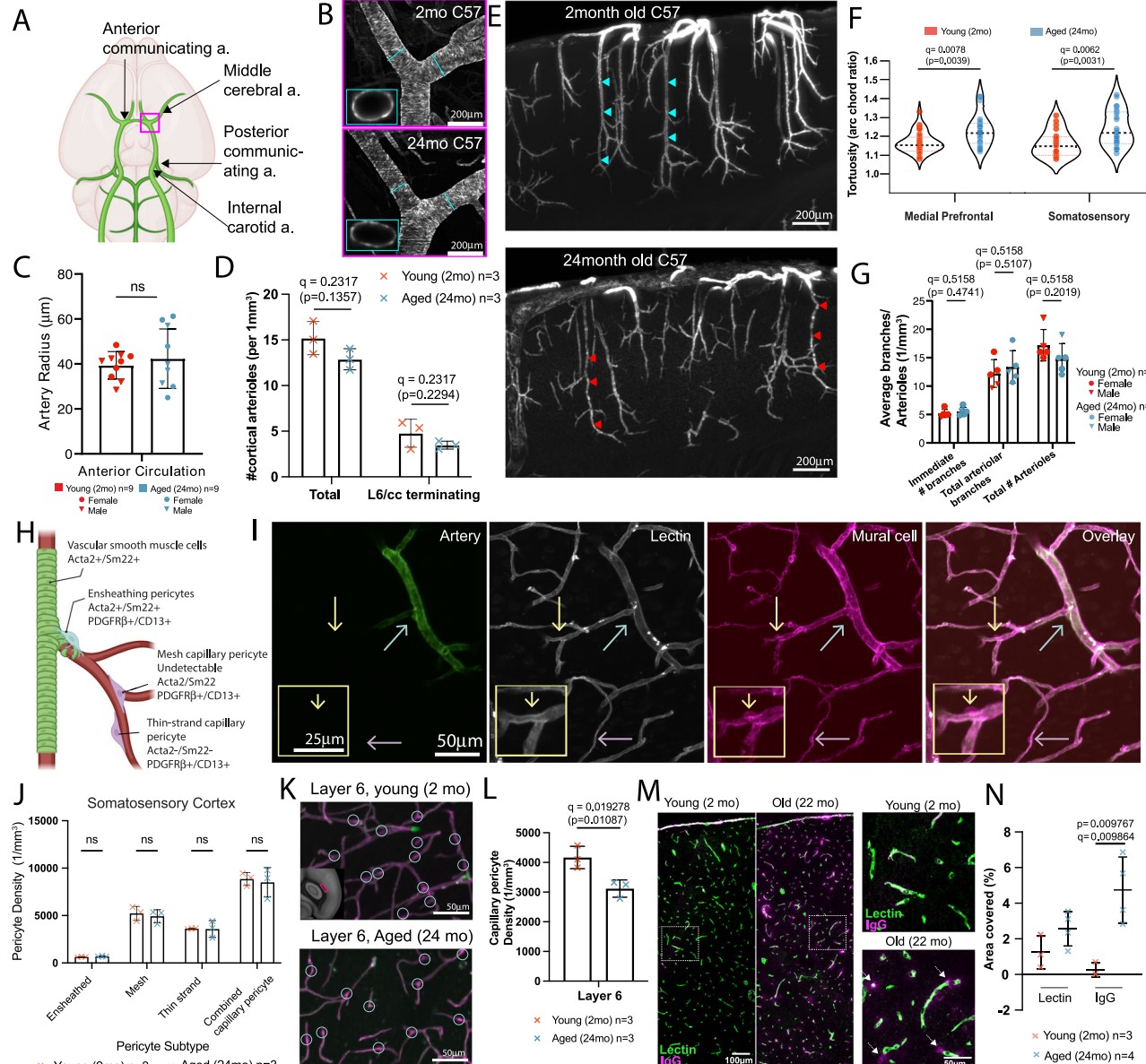

**Fig. 5 | Aging induces significant arteriole remodeling and selective pericyte density reduction. A** Schema of main arteries of the circle of Willis at the ventral surface of the brain. **B** Artery specific labeling of the middle cerebral artery branching area (red box area in H) from 2-month-old and 24-month-old brains, scale bar 200 μm. **C** Artery radii do not show a significant difference between the two age groups. **D** The number of both total and deep layer 6 reaching penetrating cortical arteriole did not show a significant difference between the two age groups. **E** Representative 600 μm MIPs of artery labeling in the somatosensory area of a young (top) and an aged (bottom) brain, scale bar 200 μm. Note tortuous arterioles in the old brain (red arrowheads) compared to straight ones in the young brain (light blue arrowheads). **F** Old brains showed significantly tortuous arterioles in the medial prefrontal and somatosensory cortices. Data from 3 animals for both young and aged groups. **G** Both immediate and total arteriole branch numbers show no significant differences between the two age groups. **H** Different pericyte subtypes with immuno markers and their position in the vascular order. **I** Submicron resolution LSFM images with artery labeling, whole vasculature labeled with lectin and mural cell labeling with PDGFRβ and CD13 antibodies. Scale bars of main images 50 μm and yellow outlined image 25 μm. The cyan arrow for an ensheathing pericyte, the yellow arrow for a mesh capillary pericyte, and the purple arrow for a thin-strand capillary pericyte. **J** Manual cell counting did not show any significant difference in the somatosensory cortex between the two age groups. **K, L** However, layer 6 of the somatosensory cortex (**K**) showed a significant reduction in pericyte density (**L**), scale bar 50 μm. **M, N** Increased IgG extravasation in 22-month-old brain compared to 2-month-old brain, scale bars of main images are 100 μm, and higher magnification images are 50 μm (**M**) and quantification (**N**). For two group comparisons, two-sided unpaired t-tests were used with multiple comparison correction to account for comparisons across multiple brain regions. For multiple comparisons two-way ANOVA, or mixed model if including NaN values, to generate comparison between groups. All q values obtained from multiple comparison correction by false discovery rate and uncorrected p-value are reported in each graph, except (**F**) with Bonferroni correction. Data are presented as mean values +/- SD in all graphs. Source data are provided as a Source Data file.

such as first order (ensheathing) pericytes at the junction between arterioles and microvessels[53]. To examine how different pericyte subtypes are differentially impacted in advanced aging, we used a combination of artery, pan-vascular, and mural cell immunolabeling to distinguish pericyte subtypes at different vascular zones with submicron resolution ($0.4 × 0.4 × 1$ μm³) using LSFM imaging (Fig. 5I). We successfully visualized individual pericytes and their subtypes, including capillary pericytes, both mesh and thin-strand morphologies, and ensheathing pericytes, which are located along pre-capillary arterioles and express smooth muscle markers such as Acta2[36]

(Fig. 5H, I). Examples of different pericyte subtypes (i.e., ensheathing, mesh, and thin strand) are labeled in Fig. 5I with cyan, yellow, and purple arrows in each panel, respectively. By following individual vasculature, each pericyte type was manually counted in a region of interest.

Consistent with our STPT data, we did not observe any significant changes in pericyte subtype density within the primary somatosensory cortex except a significant reduction of capillary pericyte density in layer 6 (Fig. 5J–L). This result further confirmed that pericyte density remains largely unchanged in sensory cortical areas, including contractile ensheathing pericytes. We further examined the entorhinal cortex, since this region is important for memory and is known to be very sensitive to age-related diseases[4,19,38,54–56]. While this region did not show any statistically significant decreases at 18 months of age (early aging) in STPT pericyte mapping (Fig. 3B), we found that both mesh and thin-strand pericytes, morphological subtypes of capillary pericytes, but not ensheathing pericytes, showed significant reductions in 24-month-old (late aging) mice (Supplementary Fig. 4). This suggests that capillary pericytes are at higher risk of cellular density loss, particularly in advanced age.

Since pericytes play a key role in regulating blood-brain barrier (BBB) properties[16,46], we examined whether aged brains show leaky BBB properties with compromised tight junction proteins. We analyzed serum protein extravasation in the somatosensory cortex (SS)[57] and found a significant increase in immunoglobulin extravasation throughout the entire layers of the SS (Fig. 5M, N), in agreement with an in vivo study[58]. However, we did not find significant changes in the ZO-1 expression, a tight junction marker, in the SS of the aged brain (Supplementary Fig. 5). This suggests that the pericyte function to maintain barrier properties is selectively compromised upon aging even without loss of their regional density.

## In vivo imaging to examine hemodynamic changes in aged brains

In addition to structural changes, the cerebrovasculature may undergo functional changes in neurovascular coupling with aging. Thus, we investigated how normal aging impacts brain hemodynamics during rest and in response to voluntary locomotion and sensory stimulation in awake-behaving mice, using wide-field intrinsic optical imaging of spectroscopy (IOS)[59] and two-photon laser scanning microscopy (2PLSM)[60–62](Figs. 6 and 7). All experiments were performed in awake mice that were head-fixed on a spherical treadmill for voluntary locomotion[59–63]. Imaging was performed through polished and reinforced thin-skull windows (PoRTS) to minimize the disruption of the intracranial environment[64]. We utilized two different models, voluntary locomotion[59,61] and whisker stimulation[60,62,65], to quantify the evoked responses. We focused our analysis on two functionally distinct cortical regions, the forelimb/hindlimb representation of the somatosensory cortex (FL/HL) and a frontal cortical region (FC), including the anterior lateral motor cortex (ALM). We targeted ALM because it is involved in motor planning and performs "higher-order" cognitive functions in mice, which makes it analogous to the human prefrontal cortex. We performed these measurements in mice of ages of 2-4 months and 18 months.

## Neurovascular coupling remains intact in 18-month-old mouse brains

We first assessed the spatial extent of cortical hemodynamic responses and their relationship to voluntary locomotion, using intrinsic optical signal imaging at multiple wavelengths[59]. Taking advantage of differences in the optical absorption spectra of oxy-hemoglobin (HbO) and deoxyhemoglobin (HbR)[66,67], we collected reflectance images during rapid alternating green (530 nm) and blue (470 nm) illumination (Fig. 6A). When the brain is illuminated with light of different wavelengths, increases in total hemoglobin

concentration (ΔHbT) in turn report dilations of arteries, capillaries, and veins, which correspond with increases in cerebral blood volume (CBV). The ΔHbT observed with IOS closely tracks measurements of vessel diameter made with two-photon microscopy[68]. The consistency of microscopic measurements of vessel diameter, combined with its very high signal-to-noise ratio[60], and spatial resolution (less than 200 μm)[69], makes IOS suitable for detecting hemodynamic responses to locomotion. While neurally-evoked dilations initiate in the deeper layers of the cortex, the dilations propagate up the vascular tree to the surface arteries[70–73], where they can be easily detected with IOS.

We quantified how locomotion affected CBV in two complementary ways. We calculated the locomotion-triggered average, generated by aligning the IOS or vessel diameter signals to the onset or offset of locomotion using only locomotion events ≥ 5 seconds in duration (Fig. 6B, C). Using changes in ΔHbT as an indicator of CBV, we observed region-specific changes in ΔHbT during locomotion (Fig. 6B, C). In young adult mice (2-4 months old), there was a pronounced increase in the ΔHbT (corresponding to an increase in CBV) in the forelimb/hindlimb representation of the somatosensory cortex (FL/HL), while in the frontal cortex (FC) there was no change, or even a slight decrease in ΔHbT ($n = 7$ mice, 4 male and 3 female) (Fig. 6B, C), consistent with previous reports[59,63,74]. This pattern was not significantly affected by aging, as we observed similar results in 18-month-old ($n = 5$ mice, all male) (Fig. 6B, C).

We also calculated the hemodynamic response function (HRF)[60,75], which is the linear kernel relating locomotion events to observed changes in CBV and vessel diameter (Fig. 6D and F; see Methods), using all locomotion events. Hemodynamic response functions are used in all of fMRI analyzes to extrapolate neuronal activity from a stimulus or a task from hemodynamic signals, and take into account the slower responses of the vasculature relative to neurons[76]. Using the HRFs to quantify the net CBV, we obtained the same conclusions as derived from the locomotion-triggered average, i.e., the net increase in cerebral blood volume does not change significantly during aging (Fig. 6E and G, left) in either FC or FL/HL (2-month: $0.53 \pm 0.18$ μM, $n = 7$ mice, 4 male and 3 female; 18-month: $0.57 \pm 0.06$ μM, $n = 5$ mice, all male). In addition to the amplitude of the hyperemic response evoked by locomotion, HRFs also provide us information regarding the temporal dynamics of CBV responses. We found that the onset time (Fig. 6E, middle, 2-month: $0.95 \pm 0.15$ s, $n = 7$ mice, 4 male and 3 female; 18-month: $0.95 \pm 0.14$ s, $n = 5$ mice, all male) and duration (Fig. 6E, right, 2-month: $1.11 \pm 0.12$ s, $n = 7$ mice, 4 male and 3 female; 18-month: $1.24 \pm 0.24$ s, $n = 5$ mice, all male) of locomotion evoked hyperemic response remained similar in aged brains. To further validate the results from HRFs, we quantified the responses of ΔHbT in response to brief whisker stimulation (100 ms duration) (Fig. 6H–L). We observed that in response to contralateral whisker stimulation, the ΔHbT response remains similar in 18-month-old brains ($n = 4$ mice, all male) compared to 2-month-old brains ($n = 5$ mice, 3 male and 2 female) (Fig. 6I–L). In addition to the mesoscopic level measurements using IOS, we further compared whether hemodynamics was different between age groups at individual vessel level in FL/HL, in terms of pial arterial diameter change in response to locomotion, using in vivo 2PLSM (Fig. 6M). The locomotion-evoked arterial diameter change (Fig. 6N), as well as the HRF of arterial diameter change (Fig. 6O) showed a similar spatial pattern of responses as the CBV measured using IOS, despite a trend of delayed response during aging.

Finally, to determine whether vascular dilation capacity was intact in aged mice, we measured the mesoscopic brain hemodynamic responses using IOS (Fig. 6P) and microscopic vessel diameter response to isoflurane, a potent vasodilator, using 2PLSM. In FL/HL, we observed an increase of ΔHbT (2-month: $176.8 \pm 29.5$ μM, 4 male mice; 18-month: $146.7 \pm 22.2$ μM, 4 male mice) and arteriole diameter (2-

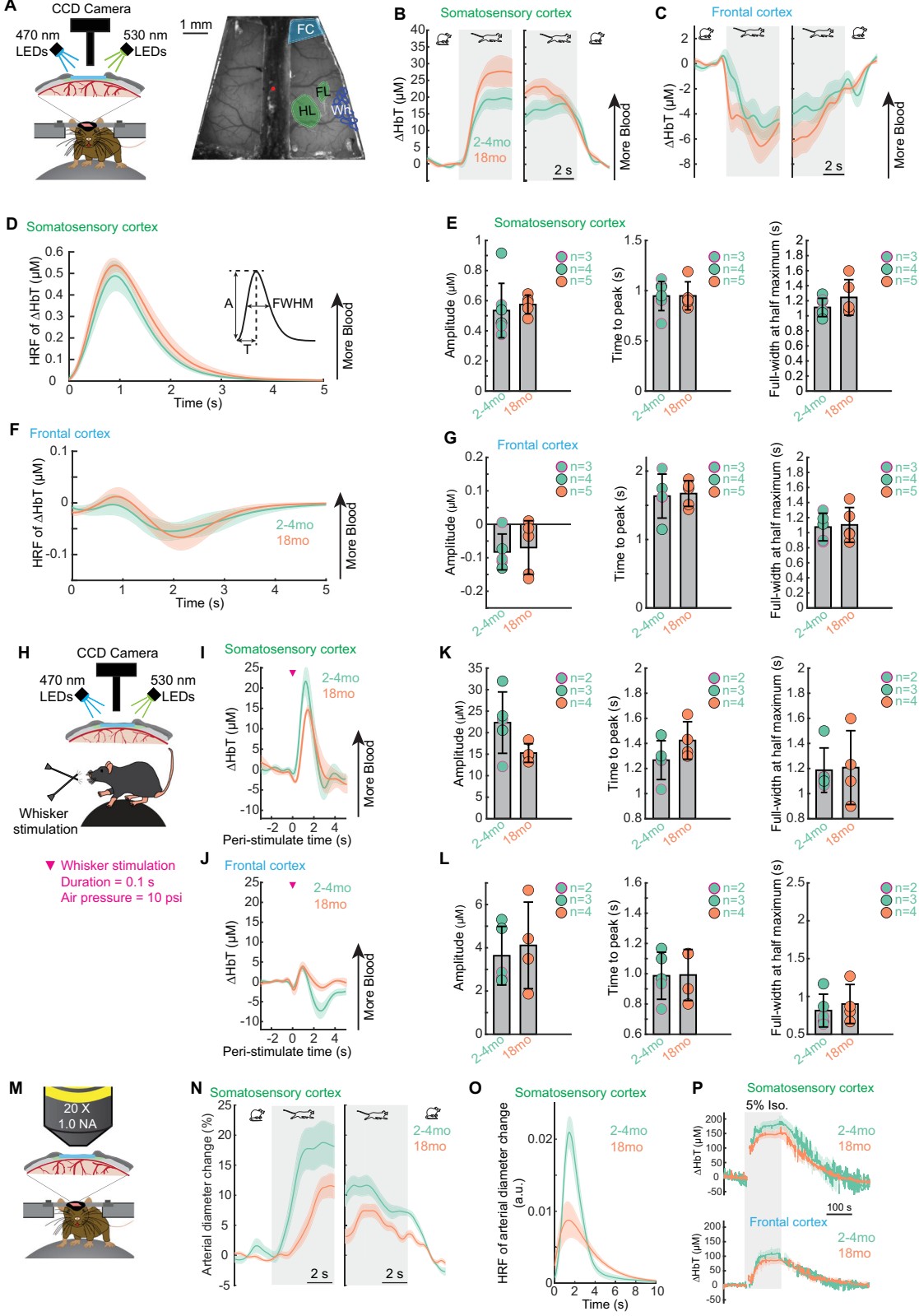

month: $65.5 \pm 19.2\%$, 4 male mice; 18-month: $51.7 \pm 10.8\%$, 5 male mice; Source Data File) under isoflurane. In FC, we observed an increase of $\Delta HbT$ (2-month: $106.6 \pm 28.5$ μM, 4 male mice; 18-month: $85.4 \pm 23.0$ μM, 4 male mice). The extent of vasodilation observed between young and old mice was not different when animals were transitioned from air to 5% isoflurane, suggesting the dilation capacity remains similar across different age groups.

**Oxygenation carrying capacity is decreased in aged mice**

One of the important functions of increased blood flow/volume is to deliver oxygen to the brain. Using the cerebral oxygenation index (HbO-HbR)[59,77], a spectroscopic measurement of hemoglobin oxygenation, we saw an increase in oxygenation during locomotion in both FC and FL/HL areas in young mice. The oxygen increase in response to locomotion (Fig. 7A) did not significantly differ across age groups. As

**Fig. 6 | Cortical hemodynamic responses to voluntary locomotion and whisker stimulation is intact in normal aging. A** Left, schematic of the experimental setup for IOS imaging during voluntary locomotion. Right, an image of thin-skull window and overlay of corresponding anatomical reconstruction; scale bar = 1 mm. FC, frontal cortex; FL/HL, forelimb/hindlimb representation of the somatosensory cortex; Wh, vibrissae cortex. **B** Population average of locomotion onset (left) and offset (right) triggered average of ΔHbT responses in FL/HL across different age groups. **C** As in (**B**) but for FC. Sample sizes for A-C include: 2-4 month old mice: n = 7 (4 males and 3 females) and 18 month old mice: n = 5 (all males). **D** Hemodynamic response function (HRF) of ΔHbT in the FL/HL across different age groups. **E** Quantification of HRF of ΔHbT in the FL/HL: amplitude (A, left), time to peak (T, middle), and full-width at half maximum (FWHM, right), for 2-4 month old: n = 7 mice (4 males and 3 females) and 18 month old mice: n = 4 (all male mice). Sex is shown with a black circle outline for males and a magenta circle outline for female mice. **F** As in (**D**) but for FC. **G** As in (**E**) but for FC. **H** Schematic of the experimental setup for IOS imaging during whisker stimulation. **I** Average population responses of ΔHbT to contralateral whisker stimulation in the FL/HL across different age groups. **J** Quantification of the whisker stimulation evoked responses of ΔHbT in the FL/HL: amplitude (left), time to peak (middle) and full-width at half maximum (right), for 2-4 month old: n = 5 mice (3 males and 2 females) and 18 month old mice: n = 4 (all male mice). Sex is shown with a black circle outline for males and a magenta circle outline for female mice. **K** As in (**I**) but for FC. **L** As in (**J**) but for FC. **M** Schematic of the experimental setup for 2PLSM imaging during locomotion. **N** Population average of locomotion onset (left) and offset (right) triggered the average of arteriole diameter responses in FL/HL across different age groups. **O** Hemodynamic response function (HRF) of arteriole diameter changes in the FL/HL across different age groups. **P** Population average of ΔHbT responses to inhalation of 5% isoflurane in the FL/HL (top) and FC (bottom) across different age groups. Sample sizes for M-P include 2-4 month-old: n = 4 (all male mice) and 18-month-old: n = 4 (all male mice). Solid lines and shaded areas in (**B, C, D, F, I, N, O, P**) denote mean ± SEM, respectively. Data are shown as mean ± SD in all other graphs (**E, G, J, L**). Source data are provided as a Source Data file.

vasodilation is one of the determining factors controlling brain oxygenation[59], we quantified the relationship between locomotion-evoked responses of ΔHbT and ΔHbO-HbR using linear regression. The slope and intercept of the fitting decreased with the healthy aging process (2-month: y = 0.8667x + 35.43, n = 7 mice, 4 male and 3 female; 18-month: y = 0.5209x + 27.37, n = 5 mice, all male; Fig. 7B), which suggests that oxygen carrying capacity for the red blood cells decreases during aging, and that the aged brain has lower baseline oxygenation, respectively.

To determine whether the oxygen exchange and oxygen delivery capacity were intact in aged mice, we measured the brain tissue oxygenation response when mice breathed 100% oxygen (Fig. 7C–F). We observed that the oxygen delivered to the brain is significantly smaller in the aged mouse brain, both in the FC (2-month: 43.7 ± 4.1 μM, n = 4 mice, 3 male and 1 female; 18-month: 32.5 ± 10.4 μM, n = 4 mice, all male. Linear mixed effects model, p = 0.0536) and FL/HL (2-month: 71.0 ± 14.4 μM, n = 4 mice, 3 male and 1 female; 18-month: 51.0 ± 5.3 μM, n = 4 mice, all male. Linear mixed effects model, p = 0.0097).

Lastly, we quantified the functions of the brain capillary network during aging progress, as its dynamics affect brain oxygenation responses[59,78]. We first compared whether red blood cell (RBC) velocity differed between age groups in the capillary network. We found no significant differences in lumen diameter between different groups (2-month: 4.7 ± 1.65 μm, 32 capillaries, n = 10 mice, 6 male and 4 female; 18-month: 4.9 ± 1.2 μm, 36 capillaries, n = 5 male mice), a trend toward decreased RBC velocity, but not a statistically significant difference (2-month: 0.58 ± 0.33 mm/s, n = 10 mice, 6 male and 4 female; 18-month: 0.53 ± 0.34 mm/s, n = 5 male mice) (Supplementary Fig. 6A), no difference in hematocrit (2-month: 38.3 ± 7.6%, n = 10 mice, 6 male and 4 female; 18-month: 33.4 ± 9.6%, n = 5 male mice; Supplementary Fig. 6B). In addition to RBC flow rate and hematocrit, the "stochastic" nature of red blood cell distribution in the capillary also affects brain oxygenation[61,78]. When we quantified the spacing of RBC and the occurrence of "stall" events, we found no significant difference between different aging groups (Supplementary Text 1).

Collectively, our in vivo recording results suggest relatively intact vascular response dynamics and decreased oxygen-carrying capacity in 18-months-old mice (early aging), which can create imbalances in baseline and on-demand supply of oxygen in aged brains.

## Discussion

Understanding structural and functional changes of the cerebrovasculature during normal aging will provide foundational information to understand altered brain energy infrastructure that can be commonly linked with many neurodegenerative disorders. Here, we provide detailed information regarding anatomical changes of the cerebrovascular network and physiological alteration of the blood flow in aged mouse brains, as summarized in Fig. 8. We found overall reductions in vascular length and branching densities with BBB impairment, along with more tortuous arterioles that indicate sparser and remodeled vascular networks in aged brains. We also uncovered selective vascular and pericyte loss in deep cortical layers, basal forebrain regions, and the hippocampal network, including the entorhinal cortex, which may contribute to their regional vulnerabilities in neurodegenerative disorders[44,79]. Lastly, our in vivo studies showed inefficient oxygen delivery in aged brains despite relatively intact neurovascular coupling response time. Collectively, our results advance our understanding of global changes and regional vulnerabilities associated with deteriorating vascular networks in aged brains.

### Cerebrovascular structural changes with selective pericyte reduction in aged brains

Previous studies in aged cerebral vasculature have shown stiffened arteries, microvascular rarefaction, and remodeled vascular trees in selected brain regions[17,20,80–82]. Our study showed that there is an approximate 10% decrease in overall vascular density, as well as branching density, in 18-month-old compared to 2-month-old mouse brains, suggesting a sparser vascular network to distribute the blood[35]. Moreover, aged brains showed substantially more tortuous penetrating arterioles, which impede blood flow by increasing flow resistance. This increase in resistance, unless countered by an increase in blood pressure, could result in reduced oxygen and nutrient supply, particularly areas distal to main arteries such as the deep cortical layers and white matter tracks[20,37,83]. Importantly, both human and murine studies have shown similar changes with tortuous vasculature, decreased vascular density, and associated slowed cerebral blood flow[20,22,81,84,85]. Such changes can lead to an increased heart rate to compensate for cerebral hypoperfusion, as frequently observed in elderly population[86]. We also found an overall increase in average vessel radii, which is largely contributed to by the microvasculature. Notably, pericytes are known to regulate the basal tone and permeability of microvessels, and release neurotropic factors[13,15,16,46]. A recent study also showed that pericytes in superficial cortical layers have impaired recovery of cellular processes in the aged brain[36]. Furthermore, we found that aged brains have a compromised BBB with increased immunoglobulin extravasation throughout cortical layers in aged brains[58]. Therefore, while pericyte cell density does not change significantly during aging, their regulatory function may be impaired, resulting in slightly dilated and leaky cerebrovasculature.

Our data showed that the vasculature of the isocortex is more resilient to aging compared to other brain regions, as evidenced by no significant changes in both microvascular and capillary pericyte densities. However, deep cortical layers, especially layer 6b, showed reduced vessel density and pericyte density, consistent with previous studies[87]. Notably, layer 6 plays a crucially important role as the

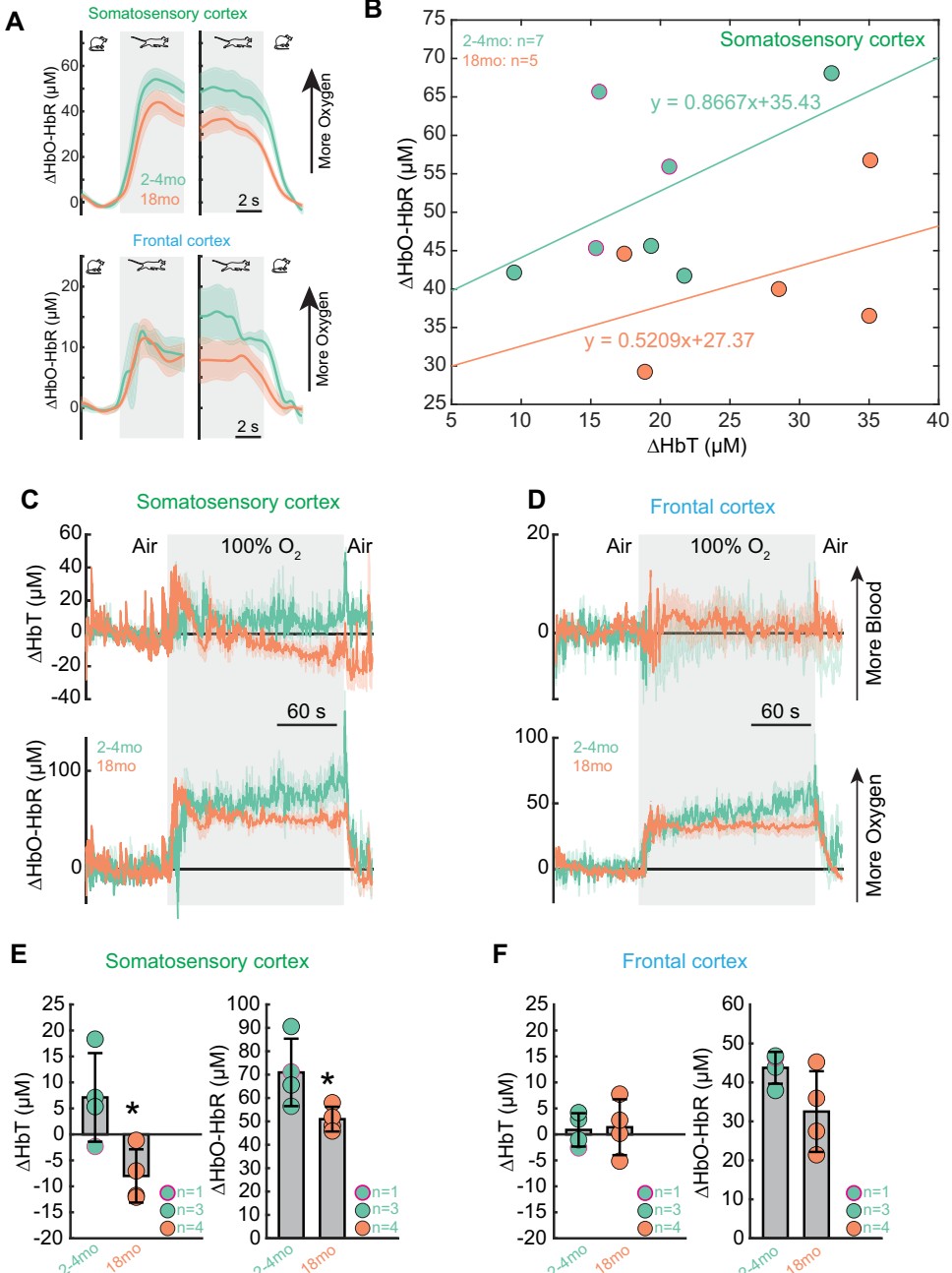

**Fig. 7 | Oxygen carrying capacity of the blood is reduced by aging. A** Population average of locomotion onset and offset triggered average of brain oxygenation (ΔHbO-HbR) responses in FL/HL and FC across different age groups. **B** Relationship between locomotion evoked change in ΔHbT and ΔHbO-HbR, 2-5 s after the onset of locomotion, across different age groups, in FL/HL, for 2month old: $n = 7$ (4 male and 3 female) and 18 month old: $n = 4$ (all male). Sex is shown with a black circle outline for males and a magenta circle outline for female mice. **C** Population average of ΔHbT (top) and ΔHbO-HbR (bottom) responses to inhalation of 100% oxygen in the FL/HL across different age groups. **D** As in (**C**) but for FC. **E** Group average of fractional changes of ΔHbT (left) and ΔHbO-HbR (right) in response to 100% oxygen in FL/HL across different age groups. For C-F, sample sizes include 2 month old: $n = 4$ (3 male and 1 female) and 18 month old: $n = 4$ (all male). Sex is shown with a black circle outline for males and a magenta circle outline for female mice. *$p$-value = 0.0097 (left), 0.0206 (right). **F** As in (**E**) but for FC. Solid lines and shaded areas in (**A**, **C**, **D**) denote mean ± SEM, respectively. Data are shown as mean ± SD in (**E**, **F**). Source data are provided as a Source Data file.

output layer to the thalamus[45,88]. Moreover, layer 6b is the only cortical layer that is responsive to sleep-wake neuropeptides such as orexin, which is produced in the lateral hypothalamus[88,89]. Considering that sleep is often dysregulated with increased age in humans[90], failing cerebrovascular network in the deep cortical layer may provide important insight to understand aging related sleep dysregulation.

Since our 3D mapping data examine vascular network changes of the whole mouse brain in an unbiased way, we identified specific brain regions with selective vulnerabilities in aged brains. For example, we found significantly reduced vascular and pericyte densities in the basal forebrain area, which contains cholinergic neurons[91]. The basal forebrain cholinergic neurons (BFCNs) have highly extensive projections to the cortical area and have large soma size with high energy demands[92]. Previous clinical and preclinical studies have shown that BFCNs are highly vulnerable in Alzheimer's disease (AD) and their deterioration is linked with memory impairment[44,93]. Impaired vascular networks with decreased pericyte density may, potentially serve as

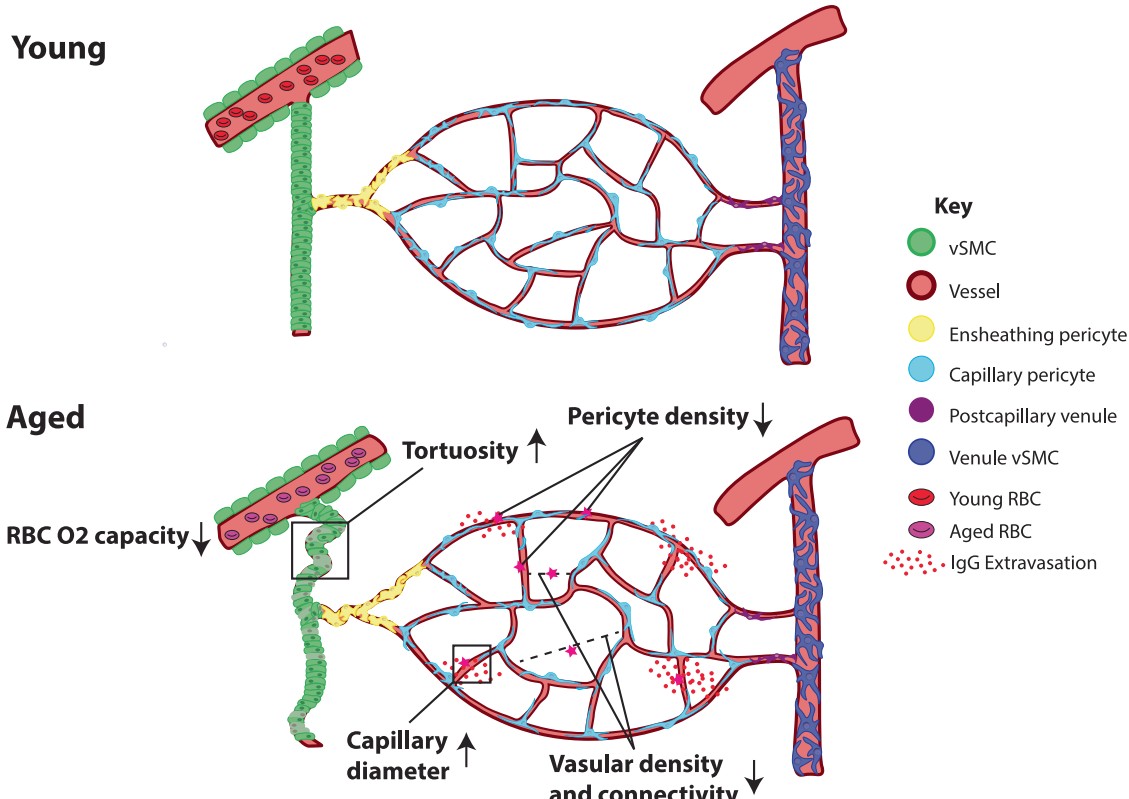

**Fig. 8 | Summary of changes in aged brains.** Aged brains show reduced vascular length and branching density, increased radii, reduced pericyte density, leaky BBB, and lower oxygen carrying capacity in the blood compared to young brains.

an underlying cause of BFCN degeneration in normal aging and neurodegenerative disorders, including AD[79,94,95]. Another notable area is the entorhinal cortex (ENT), a part of the hippocampal network, which has been heavily implicated in AD and particular cognitive deficits[96,97]. The lateral ENT (ENTl) in aged brains showed significantly decreased vascular length, branching point, and capillary pericyte density, consistent with in vivo ultrasound measurement[82]. The ENTl vascular density is one of the lowest across the brain region in normal adult mice[29]. This indicates that the entorhinal cortex is likely to be particularly susceptible to further insult (e.g., hypoxia), which may explain its vulnerability in neurodegenerative disorders. Lastly, our study identified specific thalamic and hypothalamic areas with decreases in the vascular network density, such as the medial preoptic area. This is particularly striking as the thalamus has recently been identified as a region vulnerable to microbleeds in aged mice[98]. This warrants future studies for these largely understudied subcortical areas in aging research.

**Inefficient oxygen delivery in aged brains**
We found that brain hemodynamic responses, including the increase in blood flow during locomotion and whisker stimulation in 18-month-old mice, remained relatively intact. This suggests that the aged brain can still deliver enough red blood cells to the regions with energy demands. In contrast, the baseline oxygenation and oxygen carrying capacity of the red blood cells decrease with age. Notably, respiration is an important regulator of brain oxygenation[59], and lung function decreases during the aging process[99,100]. The decreased ability to deliver oxygen can also be related to decreased microvessel density and its connectivity, resulting in less effective oxygen distribution, and the shift of the oxyhemoglobin dissociation curve with age[101]. This baseline drop in brain oxygenation will make the brain more vulnerable to hypoxia when facing increased oxygen demand, as neurons

become hyperexcitable in aged brains[102-104]. In addition, aged mice have impaired hypoxia-induced vessel formation, which is a known compensatory mechanism[105]. This baseline drop in brain oxygenation, paired with reduced vascular density, reduced vascular plasticity as well as increased blood flow resistance due to increased vessel tortuosity, will make brain areas in distal vascular territories, such as white matter tracks, and watershed areas (located at the junction between main artery territories) selectively vulnerable in aged brains[22,106]. Indeed, recent studies have demonstrated in vivo that aged animals, beginning at 18 months of age and becoming prominent at 24 months, have decreases in CBF correlated with areas of increased tortuosity in cortical areas[20,82].

We also found significant reductions in vascular and pericyte densities within the deepest cortical layers, particularly layer 6b that are likely related to the blood flow responses in both deep and superficial layers of the aged mouse cortex. A recent study utilizing 3-photon in vivo imaging demonstrated reduced vascular density with vascular network simplification and increased flow resistance in layer 6 and the corpus callosum of the somatosensory cortex, while there are increases in baseline RBC flow velocity and flux in the superficial layer[37]. In parallel, our in vivo measurement for evoked neurovascular coupling (Figs. 6 and 7) showed that functional hyperemia in the superficial layer is largely intact while RBC oxygen carrying capacity is significantly reduced. Collectively, this evidence suggests that the detrimental effects of aging occur most significantly in the deep cortical layer and may trigger potential compensatory response (e.g., increased density of pericytes) in the superficial layer, which can lead to a redistribution of blood flow toward the superficial cortical layer[37]. Coupled with less efficient on-demand oxygen delivery, these age-related changes will make deep cortical layers highly vulnerable. There is evidence that exercise can help ameliorate age-related declines in perfusion and oxygenation[107].

## Limitations of the study

In our anatomical studies, we found significant vascular loss in deep cortical areas and many subcortical areas (e.g., basal forebrains, hypothalamus, and entorhinal cortex). However, our in vivo measurement is limited to superficial cortical layers, where we did not observe dramatic anatomical changes. Although the brain hemodynamics at the surface reflect the dynamics along the vascular tree, future studies with emerging techniques such as functional ultrasound imaging or three-photon microscopy imaging will help to address functional changes in these important, yet hard-to-reach brain areas[108,109]. Moreover, another potentially informative future direction is to utilize mathematical models to delineate how vascular remodeling in aging is linked with vascular connectivity or functional impairment. Additionally, aging could affect individual animals and animals with different sexes differently. Hence, future studies that include additional behavioral and physiological measurements from both sexes with sufficient animal numbers will provide further insight to link vascular changes with individual variabilities and sex differences upon aging. Another potential limitation to this work is that the majority of C57Bl/6 aged animal groups were obtained directly from commercial vendors, which could confer differences in housing, nutrition, shipping, and handling compared to the young adult mice raised in-house. While this is unlikely to overshadow the differences due to aging, these factors could potentially impact the vasculature. Moreover, our analysis mostly focuses on the arterial and capillary compartments of the vasculature. Finally, future studies are needed to elucidate how aging affects the structure and function of the venous system of aged brains.

## Summary

Taken together, our study reveals aging-related brain-wide and area-specific changes in vascular and mural cell types. These changes can explain the vulnerability and resilience of different brain areas in normal aging, which will inform future experimental and computational approaches to gain a comprehensive understanding of brain aging. Moreover, we identified an age-related decrease in brain oxygenation and delayed neurovascular coupling, which can be linked with cognitive impairment in aged brains. These aging-related changes will serve as a common factor in understanding many neurodegenerative disorders and cognition decline in the elderly population.

## Methods

### Animals

Animal experiments were approved by the Institutional Animal Care and Use Committee at Penn State University. Adult male and female mice were used across all age and genotype groups in this study.

For transgenic pericyte-specific experiments, PDGFRβ-Cre mice (a kind gift from the Volkhard Lindner Lab)[40] were crossed with female Ai14 mice which express a Cre-dependent tdTomato fluorescent reporter (LoxP-Stop-LoxP-tdTomato). These PDGFRβ-Cre: Ai14 mice exhibit PDGFRβ expression in two distinct mural cell types, pericytes and vascular smooth muscle cells. Both the adult 2-month-old (6 males and 4 females) and 18-month-old (5 males and 4 females) PDGFRβ-Cre: Ai14 mice were bred and aged in-house. We used tail genomic DNA with PCR for the transgenic mouse lines requiring genotyping. Adult 2-month-old C57BL/6 J mice were bred from C57BL/6 J mice directly obtained from the Jackson Laboratory and used for vascular tracing experiments with FITC filling ($n = 4$)[29]. 18-month-old C57BL/6 J mice ($n = 5$) utilized for FITC-fill vascular mapping experiments were obtained directly from Jackson and Charles River laboratories. For LSFM experiments, a total of $n = 9$ Adult 2-month-old C57BL/6 J mice of both sexes (4 male and 5 female) were bred from C57BL/6 J mice directly obtained from the Jackson Laboratory and $n = 9$ 24-month-old C57BL/6 J mice of both sexes (4 male and 5 female) used for the current

study were directly obtained from the National Institute of Aging at 18 months and aged to 24 months in house. All animals were used once to generate data, and aged animals with tumors or other appreciable abnormalities were excluded from analysis.

For the in vivo two-photon imaging experiments, a total of 24 C57BL/6 J mice of both sexes (2-18 months old, 18-35 g, Jackson Laboratory) were used. Recordings of cerebral blood volume and cerebral oxygenation response to locomotion were made from 12 mice (2-4 month old: $n = 7$ mice, 4 males and 3 females; 18 month old: $n = 5$ mice, all males) using wide-field optical imaging. In a subset of the young mice (2-4-month-old: $n = 5$ mice, 3 males, and 2 females) and an additional set of aged mice (18-month-old: $n = 4$ male mice), we also recorded cerebral blood volume and cerebral oxygenation response to whisker stimulus using optical imaging. Recordings of stacks, capillary blood flow velocity, and diameters of arteries and veins using two-photon laser scanning microscopy (2PLSM) were conducted in 15 mice (12 of these 15 mice were also used for wide-field optical imaging; 2-4 month old: $n = 10$ mice, 6 males and 4 females; 18 month old: $n = 5$ male mice). Mice were given food and water *ad libitum* and maintained on 12-hour (7:00–19:00) light/dark cycles. All experiments were conducted during the light period of the cycle.

### Perfusion based vascular labeling, STPT imaging, and computational analysis

Overall procedure remains similar to our previous publication[29]. The detailed procedure has been included in a separate protocol paper[33]. Briefly, animals were deeply anesthetized with ketamine-xylazine, and perfused with 1X PBS followed by 4% paraformaldehyde to wash out blood and allow for tissue fixation, respectively. For vessel labeling, immediately following 4% paraformaldehyde, 0.1% (w/v) fluorescein isothiocyanate (FITC)-conjugated albumin (Sigma-Aldrich, cat.no.: A9771-1G) in a 2% (w/v) solution of porcine skin gelatin (Sigma-Aldrich, cat.no: G1890-500G) was perfused to obtain vascular filling. For STPT imaging, the brain sample was embedded in oxidized agarose and cross-linked in 0.05 M sodium borohydrate at 4 °C for at least 2 days ahead of imaging. We used 910 nm wavelength (UltraII, Coherent) as excitation light for all samples. Signals in the green and red spectrum were simultaneously collected using 560 nm dichroic mirror at x,y = 1,1 μm resolution in every 50 μm z (for pericyte mapping) or x,y,z = 1,1,5 μm resolution (for vascular mapping).

We utilized our previously described software pipeline to perform de-aberration, normalization, and imaging stitching steps for all STPT data collected for this study[29]. Moreover, we used the same analytical tools to binarize the vessel signals and skeletonize for further analysis. This pipeline also performs cleaning/reconnecting of artifacts, traces the vessel diameter, and finally outputs the coordinates for each vessel segment and its connectivity. The distance to the nearest vasculature is calculated by probing all tissue space in the data and finding the average distance to its nearest vasculature surface. For every point being probed, we first take all the vasculature data within the +/− 100 μm cartesian coordinated cube and calculate the straight-line distance between the probe and each vascular data point inside the cube. Then we take the minimum of those distances as the distance to the nearest vasculature for that probing point. We performed our calculation at 10 μm isotropic voxel resolution probing, which provides enough data entry for each ROI that is at least 100 μm in size. For pericyte cell counting, we used previously developed Deep Learning Neural Network (DLNN) cell counting[29]. This DLNN uses a per-cell multi-resolution-hybrid ResNet classification with potential cell locations to reduce computational time and resources without loss of quality. While aged mouse brains do have increased noise due to the accumulation of cellular debris, we validated that our DLNN pipeline performed at the same level as with young adult mice and did not incorporate cellular debris as potential cells.

## Tissue clearing, 3D immunolabeling, and LSFM imaging

Whole brain vascular staining was performed following the iDISCO+ protocol with modifications[25]. Brain samples were delipidated in SBiP buffer, consisting of ice-cold water, 50 mM Na$_2$HPO$_4$, 4% SDS, 2-methyl-2-butanol and 2-propanol. This buffer is activated at room temperature and is therefore made and stored at 4 °C before use. Each sample was submerged in 10 ml of SBiP buffer, rotated at room temperature with buffer changes at 3 hours, 6 hours and then incubated with fresh SBiP buffer overnight. For adequate delipidation, particularly for aged samples, each brain was then washed with SBiP for a total of 6 days, with daily buffer changes. After delipidation, brain samples were washed with B1n buffer, which consists of 0.1% TritonX-100, 1 g of glycine, 0.01% 10 N NaOH and 20% NaN3. Brain samples were washed with 10 ml of B1n buffer at room temperature for 2 days. To begin immunolabeling, brains were rinsed 3 times for 1 hour each with PTwH buffer, consisting of 1X PBS, 0.2% Tween-20, 10 mg heparin, and 2 g of NaN3. For primary antibody incubations, antibodies were diluted in antibody solution consisting of PTwH buffer with 5% DMSO and 3% normal donkey serum. Antibodies to smooth muscle actin (Acta2) (Rabbit anti-Acta2, Abcam, cat: ab5694, RRID:AB_2223021, dilution 1:1000) and transgelin (Sm22) (Rabbit anti-Sm22 Abcam, cat: ab14106, RRID:AB_443021, dilution 1:1500) were combined to label the artery wall[25]. Pan-vascular labeling was achieved through staining with DyLight-594 labeled Lycopersicon Esculentum (Tomato) Lectin (Vector labs, cat. no.: DL-1177-1), which was added to both primary and secondary incubations at 1:100 concentration. Pericytes were labeled by combining PDGFRβ (Goat anti- PDGFRβ, R&D Systems, cat. no.: AF1042, RRID:AB_2162633, dilution: 1:100) and Mouse Aminopeptidase N/CD13 (Goat anti-CD13, R&D Systems, cat. no.: AF2335, RRID:AB_2227288, dilution: 1:100). Primary antibodies were incubated for 10 days at 37 °C. Following primary incubation, PTwH buffer was changed 4-5 times for each sample over the course of 24 hours. A fresh antibody solution was used to dilute all secondary antibodies to a concentration of 1:500. For secondary antibodies, Alexa Fluor® 488-AffiniPure Fab Fragment Donkey Anti-Rabbit IgG (H + L) (Jackson ImmunoResearch laboratories, cat. no.: 711-547-003, RRID:AB_2340620) was used to detect artery staining and Alexa Fluor® 647-AffiniPure Fab Fragment Donkey Anti-Goat IgG (H + L) (Jackson ImmunoResearch Laboratories, cat. no.: 705-607-003, RRID:AB_2340439) was utilized to detect pericyte staining. After secondary incubation for 10 days at 37 °C, brains were washed 4-5 times in PTwH buffer for 24 hours. Brain samples were then dehydrated in a series of methanol dilutions in water (1-hour washes in 20%, 40%, 60%, 80% and 100%). An additional wash of 100% methanol was conducted overnight to remove any remaining water. The next day, brains were incubated in 66% dichloromethane/33% methanol for 3 hours and subsequently incubated in 100% dichloromethane twice for at least 15 minutes each. Brains were equilibrated in dibenzyl ether for at least two days before transitioning to ethyl cinnamate one day prior to imaging.

We used the SmartSPIM light sheet fluorescence microscope (LifeCanvas Technologies). Brains were supported in the custom sample holder by standardized pieces of dehydrated agarose consisting of 1% agarose in 1X TAE buffer. The sample holder arm was then submerged in ethyl cinnamate for imaging. We used a 3.6X objective (LifeCanvas, 0.2NA, 12 mm working distance, 1.8 µm lateral resolution) and three lasers (488 nm, 560 nm, 642 nm wavelengths) with a 2 mm step size. For detailed examination of pericytes, we used a 15X objective (Applied Scientific Instrumentation, 0.4NA, 12 mm working distance, 0.4 mm lateral resolution) with a 1µm z step size. Acquired data was stitched using custom Matlab codes adapted from Wobbly Stitcher[25].

## Analysis of LSFM-based vascular and pericyte signals

For pericyte counting, prior to quantification, each stitched image stack, per signal channel, was separately normalized, and the entire volume of each image stack was then converted to 20 µm maximum

intensity projections (MIP). Normalization of each signal channel is done by adjusting according to the histogram-determined global mean value of the background by utilizing a 10x downsized copy of the entire image stack. The 20µm MIP step was determined to prevent over or under counting of cell bodies, since pericyte cell body size tended to range from 6-10µm depending on the orientation of the cell measured within a 3D context in the original image stack. Finally, all three image channels (artery, lectin, and mural cell labels) were merged into a channel overlay to provide additional context, such as vascular zone information. Cells with stereotypical pericyte morphology (i.e., ovoid shape and protruding from the vessel wall), typically along the first through third order arteriolar branches, that also expressed smooth muscle markers and extended processes that wrapped around the vessel were classified as ensheathing pericytes. Capillary pericytes were classified according to cell body shape and localized to the capillary bed without any Acta2/Transgelin expression. These cells were further subdivided into mesh or thin strand morphologies according to their microvessel placement and type of processes, according to the definitions of these subtypes[110]. Cell bodies along larger veins, including the principle cortical venules were excluded from this analysis.

For arteriole analysis, 600 µm MIPs were obtained from the channel labeling of Acta2 and Transgelin (i.e., artery labeling). We cropped the supplementary somatosensory cortex from full datasets and quantified the total number of arteries and their branches manually.

For tortuosity measurements, a centerline of the entire vessel length was first traced to obtain the Euclidean distance (arc length) using a skeletonization tool in Clearmap 2.0[25]. Then a straight line connecting the start and end points of the previous length was obtained to measure the chord length. The arc chord ratio was then determined by dividing the Euclidean distance by the arc chord distance. We used 32 arteries from 2-month-old ($n = 3$ animals) and 23 from 24-month-old ($n = 3$ animals) in the medial prefrontal, and 19 arteries from 2-month-old ($n = 2$ animals) and 23 from 24-month-old ($n = 3$ animals). For Circle of Willis analysis, entire brain datasets for the artery channel were converted to 10µm isotropic. Next, a cropped volume including the branching point of the middle cerebral artery as well as ample segments of the anterior communicating artery and middle cerebral artery were obtained within 250 x 250 x 120 µm (x, y, z) to fully capture the entire branch point and associated arteries in x,y,z dimensions. This subset was re-sliced to obtain the cross-sectional area of this section of the vessel. The average radius was obtained from the cross-sectional areas.

For length density measurement, we modified TubeMap vascular tracing codes[25]. We devised an ilastik machine learning-based vascular detection method using all three imaging channels followed by binarization and skeletonization. We used TubeMap graph tracing tool to follow vasculature continuously to calculate vascular length in 3D volume. Moreover, we used Elastix based image registration to calculate volumes of different brain regions and to map signals in the 3D reference brain.

To compare the vasculature between in vivo two-photon and LSFM imaging from the same brain, we found a matched imaging window based on vascular architecture and overall brain anatomy in both imaging. We conducted vessel diameter measurements in 2D with vasometric (ImageJ) that allows for line measurements of diameter at every 2 µm intervals.

We initially acquired LSFM data with vascular labeling ($n = 9$ each for 2 months and 24 months old) and used them for Circle of Willis analysis. We used a subset of samples randomly chosen from data with satisfactory immuno-staining qualities of target structures.

## Immunohistochemistry

The mice were deeply anesthetized using isoflurane and then decapitated with scissors. The brain was immediately extracted and

submerged in optimal cutting temperature medium (Tissue-Tek). Rapid freezing of the immersed brain was achieved by exposure to dry ice-chilled 2-methylbutane. This frozen brain tissue was subsequently stored at −80 °C until needed. Coronal brain sections, each 10 µm thick, were obtained using a cryostat. Once brought to room temperature, the sections were fixed using a 4% paraformaldehyde solution. After fixation, the sections underwent a sequence of steps: they were rinsed three times in 1X PBS and then blocked for an hour at room temperature using 1% donkey serum diluted in PBST (1x PBS + 0.03% Triton-X). Following blocking, the slices were incubated overnight at 4 °C in a primary antibody solution (polyclonal rabbit anti-zo-1, Invitrogen, Cat# 40-2200, RRID: AB_2533456, diluted 1:200) in the blocking buffer with gentle rotation. Post-primary antibody incubation, the slices were washed three times in 1X PBS and then incubated for 1 h at room temperature with a secondary antibody (Donkey anti-rabbit conjugated with Alexa 568, Thermo Fisher Scientific, Cat# A10042, RRID: AB_2534017, diluted 1:500) along with DyLight-488 labeled Lycopersicon Esculentum (Tomato) Lectin (Vector Laboratories, Cat# DL-1174, RRID: AB_2336404, dilution 1:100). Prior to mounting, the slices were washed three times in 1X PBS and then mounted using vectashield mounting media containing DAPI (Vector Laboratories, Cat# H-1500, RRID: AB_2336788).

For IgG staining, the procedure was used for fixing and blocking. Subsequently, the sections were incubated for 20 hours at 4 °C with anti-IgG (CF640R, Biotium, Cat# 20177, RRID: AB_10853475, dilution 1:100) in blocking buffer. Following this incubation, the samples were washed three times with 1X PBS and incubated with DyLight-488 labeled Lycopersicon Esculentum (Tomato) Lectin (Vector Laboratories Cat# DL-1174, RRID: AB_2336404, dilution 1:100) for 1 h at room temperature. After three washes with 1X PBS, the sections were mounted using vectashield mounting media containing DAPI (Vector Laboratories Cat# H-1500, RRID: AB_2336788).

Images were acquired using a confocal microscope (Zeiss LSM900 with Airyscan 2) using a 20x objective. The images were then binarized and area coverage was calculated using 'analyze particles' function in ImageJ. All the images for the same staining were imaged under the same settings and were analyzed using the same binarizing parameters.

### Surgery, habituation, and measurement for in vivo recording

Cerebral oxygenation, cerebral blood volume (CBV) and vessel diameter data were acquired from the same groups of awake, behaving mice during voluntary locomotion and whisker stimulation. All surgeries were performed under isoflurane anesthesia (in oxygen, 5% for induction and 1.5-2% for maintenance). A custom-machined titanium head bolt was attached to the skull with cyanoacrylate glue (#32002, Vibra-tite). The head bolt was positioned along the midline and just posterior to the lambda cranial suture. Two self-tapping 3/32" #000 screws (J.I. Morris) were implanted into the skull contralateral to the measurement sites over the frontal lobe and parietal lobe. For measurements using two-photon laser scanning microscopy (2PLSM), CBV measurement using intrinsic optical signal (IOS) imaging or brain oxygenation measurement using spectroscopy, a polished and reinforced thin-skull (PoRTS) window was made covering the right hemisphere or both hemispheres[59–64,68]. Following the surgery, mice were then returned to their home cage for recovery for at least one week, and then started habituation on experimental apparatus. Habituation sessions were performed 2-4 times per day over the course of one week, with the duration increasing from 5 min to 45 min.

Habituation: Animals were gradually acclimated to head-fixation on a spherical treadmill[59,61,63,111] with one degree of freedom over at least three habituation sessions. The spherical treadmill was covered with nonabrasive anti-slip tape (McMaster-Carr) and attached to an optical rotary encoder (#E7PD-720-118, US Digital) to monitor locomotion. Mice were acclimated to head-fixation for ~15 minutes during the first session and were head-fixed for longer durations (>1 hour) in the subsequent sessions. Mice were monitored for any signs of stress during habituation. In all cases, the mice exhibited normal behaviors such as exploratory whisking and occasional grooming after being head-fixed. Heart rate fluctuations were detectable in the intrinsic optical signal[59,68] and varied between 7 and 13 Hz for all mice after habituation, which is comparable to the mean heart rate (~12 Hz) recorded telemetrically from mice in their home cage[112]. Habituation sessions were achieved 2-4 times per day over the course of one week, with the duration increasing from 5 min to 45 min. Mice that received whisker stimulation ($n = 9$) were acclimatized to head-fixation for 15–30 min during the first session. In subsequent sessions, they began to receive air puffs directed at the whiskers and were head-fixed for longer durations (>60 minutes).

Physiological measurements: Data from all experiments (except two photon laser scanning microscopy) were collected using custom software written in LabVIEW (version 2014, National Instruments). We focused on two different brain areas, the somatosensory cortex (the forelimb/hindlimb area, FL/HL) and a frontal cortical region (FC), including the anterior lateral motor cortex (ALM). The coordinates are as follows: 0-1 mm caudal and 1-2 mm lateral from bregma, ~1 mm² for FL/HL and 2–4 mm rostral and 0.5–2.5 mm lateral from bregma, ~4mm² for the ALM.

Behavioral measurement: The treadmill movements were used to quantify the locomotion events of the mouse. The animal was also monitored using a webcam (Microsoft LifeCam Cinema®) as an additional behavioral measurement.

Vibrissa stimulation: Animals were awake and engaged in whisking behavior during IOS data acquisition[60,62]. Brief (0.1-s duration) puffs of air were delivered to the ipsilateral and contralateral whiskers through a thin plastic tube (length 130 mm, diameter 2 mm). Air puffs were directed to the distal ends of the whiskers at an angle parallel to the face to prevent stimulation of other parts of the head or face. An additional air puffer was set up to point away from the body for use as an auditory stimulus. The puffs were delivered via solenoid actuator valves (Sizto Tech Corporation, 2V025 1/4) at constant air pressure (10 psi) maintained by an upstream regulator (Wilkerson, R03-02-000). Air puffs were separated by intervals of 30-60 s, and the order of all sensory stimulation was randomized, with a nominal ratio of three contralateral stimuli for every ipsilateral or auditory stimulation. Auditory and ipsilateral stimuli were omitted from the principal analysis because their responses were primarily related to stimulus-provoked movement.

Brain oxygen measurement using optical imaging: We mapped the spatiotemporal dynamics of oxyhemoglobin and deoxyhemoglobin concentrations using their oxygen-dependent optical absorption spectra[78]. Reflectance images were collected during periods of green LED light illumination at 530 nm (equally absorbed by oxygenated and deoxygenated hemoglobin, M530L3, Thorlabs) or blue LED light illumination at 470 nm (absorbed more by oxygenated than deoxygenated hemoglobin, M470L3, Thorlabs). For these experiments, a CCD camera (Dalsa 1M60) was operated at 60 Hz with 4×4 binning (256 × 256 pixels), mounted with a VZM300i optical zoom lens (Edmund Optics). Green and blue reflectance data were converted to changes in oxy- and deoxyhemoglobin concentrations using the modified Beer-Lambert law with Monte Carlo-derived wavelength-dependent path length factors[67]. We used the cerebral oxygenation index[77] (i.e., HbO-HbR) to quantify the change in oxygenation, as calculating the percentage change requires knowledge of the concentration of hemoglobin on a pixel-by-pixel basis, which is not feasible given the wide heterogeneity in the density of the cortical vasculature[32].

Measurements using two-photon laser scanning microscopy (2PLSM): Mice were briefly anesthetized with isoflurane (5% in oxygen)

and retro-orbitally injected with 50 μL 5% (weight/volume in saline) fluorescein-conjugated dextran (70 kDa, Sigma-Aldrich), and then fixed on a spherical treadmill. Imaging was done on a Sutter Movable Objective Microscope with a 20X, 1.0 NA water dipping objective (Olympus, XLUMPlanFLN). A MaiTai HP (Spectra-Physics, Santa Clara, CA) laser tuned to 800 nm was used for fluorophore excitation. All imaging with the water-immersion lens was done with room temperature distilled water. All the 2PLSM measurements were started at least 20 minutes after isoflurane exposure to avoid the disruption of physiological signals due to anesthetics.

For navigational purposes, wide field images were collected to generate vascular maps of brain pial vascular maps of the entire PoRTS window. We performed three different measurements using 2PLSM. (1) To measure blood vessel diameter responses to locomotion, individual arteries and veins were imaged at nominal frame rate of 3 Hz for 5 minutes using 10-15 mW of power exiting the objective. Diameter of pial vessels were calculated[65]. (2) To measure RBC velocity and RBC spacing, line scan images were collected from individual capillaries (diameter range: 2–8 μm). The pixel dwell time for the line scan segments was 1 μs and we achieved a ~ 1.5 kHz sampling rate. (3) To measure the vasculature diameter under physiological conditions (i.e., awake and resting), we collected stack image every other day for each mouse. For each mouse, we collected data from 4 different days and collected 3 different trials on each day. Shortly (within 20 minutes) after the last trial on the last day, the mouse was perfused for future vasculature reconstruction. The resolution for each XY plane is 0.64 μm/pixel and the resolution for Z direction is 1 μm. On the Z-direction, three frames were collected and averaged, the averaged frame was saved in the file. All the images were acquired with increasing laser power up to 100 mW at a depth of ~200 um.

Isoflurane challenge: To compare the capability of vasodilation in both young and aged mice, we exposed a subset of mice to short period ( ~ 2 minutes) of isoflurane (5% in pure oxygen) and imaged the pial vessel (specifically, the branch of the middle cerebral artery) diameter responses. This allowed us to assess the magnitude of diameter change of pial arteries and veins.

Oxygen challenge experiments: In a subset of experiments, hyperoxia was induced by substituting breathing air for 100% pure oxygen. Using optical imaging of spectroscopy, we performed an oxygen challenge. Mice were head-fixed on a spherical treadmill, and a nose cone was fixed ~ 1 inch in front of the nose, with care taken not to contact the whiskers. Two gases were administered during a 5-min spectroscopy trial in the following order: 1 min breathable air (21% oxygen), 3 min 100% oxygen, and 1 min breathable air. Mice breathed breathable air for at least 2 min between trials, to ensure physiological parameters returned to baseline. Reflectance images were collected during periods of green LED light illumination at 530 nm (equally absorbed by oxygenated and deoxygenated hemoglobin, M530L3, Thorlabs) or blue LED light illumination at 470 nm (absorbed more by oxygenated than deoxygenated hemoglobin, M470L3, Thorlabs) or red LED light illumination at 660 nm (absorbed more by deoxygenated than oxygenated hemoglobin, M660L2, Thorlabs).

## Data analysis for in vivo recording
All data analyzes were performed in Matlab (R2019b, MathWorks) using custom code.

Locomotion event identification: Locomotion events[59,63,111] from the spherical treadmill were identified by first applying a low-pass filter (10 Hz, 5th order Butterworth) to the velocity signal from the optical rotary encoder, and then comparing the absolute value of acceleration (first derivative of the velocity signal) to a threshold of 3 cm/s². Periods of locomotion were categorized based on the binarized detection of

the treadmill acceleration:

$$\delta(t) = \theta(|a_t| - a_c) = \begin{cases} 1, & |a_t| \geq a_c \\ 0, & |a_t| < a_c \end{cases} \tag{1}$$

where $a_t$ is the acceleration at time t, and $a_c$ is the treadmill acceleration threshold.

Spontaneous activity: To characterize spontaneous (non-locomotion-evoked) activity, we defined "resting" periods as periods started 4 seconds after the end of previous locomotion event and lasting no less than 60 seconds.

Calculation of hemodynamic response function: We considered the neurovascular relationship to be a linear time invariant system[75,113,114]. To provide a model-free approach to assess the relationship between CBV or vessel diameter and neural activity, hemodynamic response function (HRF) was calculated by deconvoluting CBV signal, oxygen signal or vessel diameter signal to locomotion events, respectively, using the following equation:

$$H_{(k+1) \times 1} = \left(L^T L\right)^{-1} L^T V_{(m+k) \times 1} \tag{2}$$

H is the HRF, V is the tissue oxygenation signal or neural activity signal. L is a Toeplitz matrix of size (m + k) x (k + 1) containing binarized locomotion events (n):

$$L(\vec{n}) = \begin{pmatrix} 1 & n_1 & 0 & 0 & \cdots & 0 \\ 1 & n_2 & n_1 & 0 & \cdots & 0 \\ \vdots & \vdots & n_2 & n_1 & \cdots & \vdots \\ \vdots & n_k & \vdots & n_2 & \cdots & n_1 \\ \vdots & 0 & n_k & \vdots & \cdots & n_2 \\ \vdots & \vdots & \vdots & n_k & \ddots & \vdots \\ 1 & 0 & 0 & 0 & \cdots & n_k \end{pmatrix} \tag{3}$$

Comparison of HRF parameters: To quantify the temporal features of HRF, the HRF for CBV was fitted using a gamma-variate fitting process[75,115–118] using a gamma-variate function kernel of the following form,

$$HRF(t,T,W,A) = A * \left(\frac{t}{T}\right)^{\alpha} * e^{\left(\frac{t-T}{\beta}\right)}, \tag{4}$$

where (5) $\alpha = (T/W)^2 * 8.0 * \log(2.0)$, $\beta = W^2/(T * 8.0 * \log(2.0))$. For modeling HRF using a gamma-variate function kernel, we used a downhill simplex algorithm minimizing the sum square difference between measured and predicted hemodynamics. The goodness of fit was quantified as (6) $R^2 = 1 - \frac{\sum (HRF_{actual} - HRF_{model})^2}{\sum (HRF_{actual} - \overline{HRF})^2}$, where $\overline{HRF}$ is the mean value of the actual HRF. To quantify the amplitude of each HRF, we used the value at the peak of the modeled HRF. Time to peak (TTP) was calculated as the time at which the modeled HRF reached its maximum amplitude. Full-width at half maximum (FWHM) was defined as the time from which the modeled HRF rose to 50% of its peak until it fell to 50% of its peak. TTP, FWHM and HRF amplitudes across different cortical depths were compared using a linear model to quantify trends (robustfit, MATLAB).

2PLSM image processing: (1) To quantify blood vessel diameter responses to locomotion, individual frames from 2PLSM imaging were aligned using a rigid registration algorithm to remove motion artifacts in the x−y plane[65]. Visual inspection of movies indicated that

there was minimal z-axis motion. A rectangular box was manually drawn around a short segment of the vessel and the pixel intensity was averaged along the long axis[65]. Pixel intensity was used to calculate diameter from the full-width at half-maximum. Periods of rest were segregated using locomotion events measured with the rotary encoder. For each 5-min trial, diameter measurements were normalized to the average diameter during periods of rest. The diameters were smoothed with a third-order, 15-point Savitzky–Golay filter (Matlab function: sgolayfilt). (2) To quantify RBC velocity, blood flow velocity was calculated using Radon transform[64]. Only blood flow velocity during resting periods was reported. Capillary diameter was manually measured using ImageJ software. To quantify RBC spacing, we utilized the method reported in our previous study[61]. We identified RBC "stall" events as an inter-RBC spacing greater than 1 second. We only used RBCs spacing intervals during relatively long resting segments (i.e., ≥ 5 second). (3) As the perfusion procedure and brain fixation might affect the brain vasculature[119], to compare our measurements for vessel radii in STPT and LSFM imaging datasets to vessel parameters measured in vivo using 2PLSM, the same animals that were used for 2PLSM and STPT imaging were reconstructed and compared[29].

### Statistics and reproducibility

For the STPT and LSFM datasets, we used Matlab (Mathworks) and/or Prism (Graphpad) for all statistical analysis, including multi-region of interest (ROI) correlation analysis. Data was organized in Prism using grouped analysis with each individual animal included in subcolumns within each age group. The grouped analysis also allowed for comparisons between brain regions, with each brain region included as a row replicate. All data were reported as the mean ± standard deviation (SD), while treating each anatomical subregion (ROI) as an individual data point. For two group comparisons, multiple unpaired t-tests were used with multiple comparison corrections to correct for comparisons across multiple brain regions. The p-value was adjusted with the false discovery rate for multiple comparison corrections using the Two-stage step-up method of Benjamini, Krieger and Yekutieli in Graphpad. For multiple group comparisons, two-way ANOVA, or mixed model if including NaN values, to generate comparison between groups using Prism. Scatter plots were generated with Matlab, all other graphs were generated with Prism version 9 software.

For in vivo recording, all summary data were reported as the mean ± standard deviation (SD) unless stated otherwise. The normality of the samples was tested before statistical testing using the Anderson-Darling test (adtest). For comparison of multiple populations, the assumption of equal variance for parametric statistical method was also tested (vartest2). If criteria of normality and equal variance were not met, parametric tests (unpaired t test) were replaced with a nonparametric method (Wilcoxon rank sum test). For comparisons of oxygen challenge and isoflurane challenge effects on brain hemodynamics across different age groups, we used the linear mixed effect model (MATLAB function: fitlme). Significance was accepted at $p < 0.05$.

We used representative micrographs in Figs. 2L, M, 3D, 4D–G, 5I. We looked through all samples to choose representative images with high-quality tracing and cell counting. We confirmed we saw similar results in other samples.

### Lead contact

Further information and requests for resources should be directed to and will be fulfilled by the lead contact, Yongsoo Kim (yuk17@psu.edu).

### Reporting summary

Further information on research design is available in the Nature Portfolio Reporting Summary linked to this article.

## Data availability

Data supporting the findings described in this manuscript are available in the article, in the Supplementary Information and from the corresponding author upon request. Source data are provided with the paper. All datasets can be used for non-profit research without any restriction. The cell density and vessel metrics data used to generate figures in this study are provided in the Source Data file. The whole brain imaging data including STPT full-resolution images used in this study are available in the Brain Image Library database at https://doi.org/10.35077/g.1158. Source data are provided with this paper.

## Code availability

All codes can be used for non-profit research without any restriction. Code pertaining to STPT-related analysis, including STPT imaging reconstruction, machine learning-based cell counting, vascular tracing algorithm based on STPT imaging, cortical flatmap and elastix is previously published[29]. Our vascular tracing postprocessing and nearest neighborhood distance codes can be accessed at https://doi.org/10.5281/zenodo.11632487 and https://doi.org/10.5281/zenodo.11632393.

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

## Acknowledgements

We thank Josephine Liwang for constructive discussion of the manuscript, Dr. Volkhard Linder for kindly sharing PDGFRβ-Cre transgenic mice and Dr. Zhuhao Wu for sharing tissue clearing protocol. We thank Jen Minteer and Becca Betty for valuable management of our animal colony, as well as the coding support provided by Fae Kronman. We acknowledge the use of computational resources in the High Performance Computing cluster at the Penn State College of Medicine. Figures 1, 4A, 5A, and 5H were created with BioRender.com and released under a Creative Commons Attribution-NonCommercial-NoDerivs 4.0 International license. The work has been funded by National Institutes of Health (NIH) grant R01NS108407 and RF1MH12460501 to YK, NIH R01NS078168 and R01NS101353 to PJD, American Heart Association Career Development Award #935961 to QZ. Its contents are solely the responsibility of the authors and do not necessarily represent the views of the funding agency.

## Author contributions

Conceptualization: Y.K. and H.C.B. Anatomical Data Collection: H.C.B., Q.Z., S.B., U.C., and Y.K. Developing Computational Analysis: Y.T.W. and D.J.V. Data Analysis: H.C.B., Q.Z., Y.T.W., S.B., D.J.V., D.S., H.P., and Y.K. In vivo Imaging and related analysis: Q.Z. and P.J.D. Manuscript preparation: H.C.B., Y.K., Q.Z., and S.B. with help from other authors.

## Competing interests

The authors declare no competing interests.
