## [Peer Review File · Nature Communications]

Aging drives cerebrovascular network remodeling and functional changes in the mouse brainREVIEWER COMMENTS

Reviewer #1 (Remarks to the Author):

In this manuscript by Bennett et. Al., the authors utilized the vascular component labeling and mesoscale imaging techniques to investigate the changes in vessels and vascular cell components, as well as the blood oxygenation in the cortex during the aging process. Although still left many questions unsolved, this paper is valuable work in vascular aging research, but the paper has some shortcomings in its present form. I have several major critiques and a few minor suggestions.

- 1) The authors should describe in more details how to define the early aging and late aging in mice? As they mentioned in the manuscript, the 18 and 24-months old mice were in early and late aging respectively. Did the authors consider to utilize other structural/physiological indicators to determine the aging state for each animal?
- 2) The analysis of the changes in vascular density and capillary pericytes primarily utilized 18-month-old mice, while the examination of other vascular components and mural cells primarily utilized 24-month-old mice. What is the contribution of the vessel structural alterations to the changes in physiological parameters during late aging? The vessel structural data of 24-month-old mice needs to be provided.
- 3) The vascular parameters and components, such as artery radius and vascular tortuosity, were quantified using 3D immunostaining and tissue clearing in late aging mice. The tissue deformation rate, especially the vascular deformation rate for different aged samples, should be taken into consideration.
- 4) An interesting point that needs to be discussed, Fig. 2C shows that the volume of the isocortex shrinks in aged mice, which is a vulnerable region in aging. However, the degeneration of vessels occurred only in layer 6 (Fig. 2K-N), especially layer 6b nearby white matter. The pericyte density showed decrease in layer 6b too (Fig. 3), while the layers 2/3 and 4 showed significant increases. But the in vivo imaging only recorded superficial vascular signals. Which structural change has a greater impact? Or what is the impact of changes in layers 2 and 3 on the blood pressure and other parameters?
- 5) It's more interesting to couple the physiological changes and vessel structural changes in the same mouse, for further modeling the effect of vascular structural changes on vascular functions. In line 325 and 327-329, the authors mentioned that "the aging brain has lower baseline oxygenation" "We observed that the oxygen delivered to the brain is significantly smaller in the aged mouse brain". However, in Fig. 7F, the ΔHbT parameter in somatosensory cortex of 24-month-old mice was higher than that of 18-month-old mice. How many animals were used in these experiments (Fig. 6 and Fig. 7)? Were they also analyzed in Fig. 7 C and D? Moreover, did the authors use the same animals in Fig. 6 and Fig. 7?
- 6) In terms of age group, aged mice were divided into early old age (18-month-old) and late aging (24-month-old). However, in whisker stimulation experiment, the data of 18-month-old is lacking (Fig 6).
- 7) In Fig. 1 "3-pericyte cell counting", the signal detection of FITC with binarization appears to be positional bias.
- 8) For "in vivo imaging", forelimb/hindlimb part of somatosensory cortex and anterior lateral motor cortex are chosen as the targeted area. There were differences between the results of two areas. Did the location of the observation window in the experiment affect the results? The authors should specify the relevant parameters in the methods.
- 9) In Fig 5I LSFM images, immunostaining lacks cell nucleus localization information of mural cells.

Reviewer #2 (Remarks to the Author):

Review:

Aging drives cerebrovascular network remodeling and functional changes in the mouse brain

Authors:

Hannah C. Bennett, Qingguang Zhang, Yuan-ting Wu, Uree Chon, Hyun-Jae Pi, Patrick J. Drew, Yongsoo Kim

Summary:

The authors use mesoscale microscopy to determine vascular length and pericyte density changes at different brain regions in aged mice. They further show changes in hemodynamic responses to

sensorimotor stimulations using two photon-imaging in awake aged mice.

Comments:

1. Authors are encouraged to expand on the novelty aspects of the manuscript. For example, it would be great to see a discussion how this work advances or diverges from previous studies that performed 3D brain vascular imaging in adult and aged mice (e.g., PMID: 30007164, PMID: 23093067, PMID: 32641073, PMID: 35022482)

2. Vascular readouts: Additionally, to the vascular length and branching density it may be interesting to look at the nearest neighbor distance (NND) and distribution of between the individual vessels. These parameters have been shown to be important determining regional hypoxic areas (PMID: 26567707, PMID: 32265643)

3. Two-month old mice seem more of an adolescent age than fully adult stage of the mice. Usually, mice from 3 months are considered adult (e.g., according to JAX website). Also, some of the aged mice were 18 months that were used in some experiments; this may represent an age equivalent to a 55 year old human; that should be further discussed.

4. Both sexes of animals were used but analysed only in a subset of the experiments (Fig. 5). However, it is known that differences in vascular anatomy and CBF exist in mice and human (e.g., PMID: 27629099; PMID: 27844273). Did authors observe any region specific changes in vascular length/branching in the analysis Fig. 2-3?

5. Did authors observe any leakage of the vascular tracer in aged mice? (PMID: 32641073). In my view, it is important to understand if the described selective reduction of pericyte density (reported e.g., in Fig. 3 and Fig. 5) leads to functional effects such as increased BBB leakage. Did the authors look at presence of tight junction proteins in these areas?

6. Several studies have previously shown that reduced pericyte coverage leads to BBB leaks. Authors should consider to cite and discussed these studies (e.g., PMID: 21040844, PMID: 31235908)

7. Authors claim an increase of total brain volume by 6%, is this result statistically significant? How many animals were used and what was the standard deviation? The visualization in Fig 2B,C does not provide the information. Same goes for Fig 2D-F; is the decrease 10-20% statistically significant? In general, more information about the statistical analysis in Fig. 2,3 would be helpful. Did the authors correct their statistical method for each tested brain region?

8. It would be interesting to know if the aging-induced remodeling in arterioles are associated with differences in functional readouts such as CBF (Fig.5). Did authors perform any CBF readouts of whole mouse brains? Such as Laser Speckle/Doppler Imaging/fMRI?

Minor

1. The plots in Fig 5 C,G, J, N are too small
2. Statistical section should be explained in more detail and significance should be indicated consistently in all Fig panels.

Reviewer #3 (Remarks to the Author):

This is a comprehensive assessment of brain-wide cerebrovascular structural and functional changes with normal aging in mice. The authors assess multiple parameters of cerebrovascular structure at three ages (2, 18 and 24 months of age) and do so in both sexes on mice. They do a fair (in most cases) assessment of their results and compare these findings to those published for individual brain regions. In brief, this is a comprehensive assessment that has never before been reported.

A few weaknesses are noted below:

1. Both sexes are included in all studies, but on only one occasion are sexes separated in presentation of the data and/or discussed in interpretation of the results. In the one case where sex of mice was shown, there are too few animals in each group to test for sex differences.

2. The authors use the terms "early aging" (18 months) and "late aging" (24 month), without a definition of these terms. How physiologically or pathophysiologically are mice at these ages different? A statement about where these two ages are relative to the median life space of the mice strain would be appropriate.

3. In studies where a subset of mice was selected for assessments from a larger group under investigation, how was this selection made? This is important as it needs to be done randomly and irrespective of the data from the larger group. This selection process should be stated in the manuscript.

4. The authors state "Adult 2-month-old C57BL/6J mice were bred from C57BL/6J mice directly obtained from the Jackson Laboratory and used for vascular tracing experiments with FITC filling (n=4). 18-month-old C57BL/6J mice utilized for FITC-fill vascular mapping experiments were aged from a local C57BL/6 mouse colony. 24-month-old C57BL/6J mice used for the current study were directly obtained from the National Institute of Aging at 18 months and aged to 24 months in house." As such, it appears that each age group may have had a difference housing experience (housing environment like lighting and food source), shipping and handling. Could these factor contribute to the apparent aging effects reported? The authors failed to mention this in the potential limitation of the study.

REVIEWER COMMENTS

To all reviewers;

The removal of the 24-month-old in vivo recording data does not significantly change the overall claim of the manuscript. Rather, it helps us to focus on our findings on changes in early aging. This change has no impact in the anatomical mapping parts in Fig. 1-5. The removal of the 24-month-old in vivo recording data does not significantly change the overall claim of the manuscript. Rather, it helps us to focus on our findings on changes in early aging. This change has no impact in the anatomical mapping parts in Fig. 1-5.

Reviewer #1 (Remarks to the Author):

In this manuscript by Bennett et. Al., the authors utilized the vascular component labeling and mesoscale imaging techniques to investigate the changes in vessels and vascular cell components, as well as the blood oxygenation in the cortex during the aging process. Although still left many questions unsolved, this paper is valuable work in vascular aging research, but the paper has some shortcomings in its present form. I have several major critiques and a few minor suggestions.

1) The authors should describe in more details how to define the early aging and late aging in mice? As they mentioned in the manuscript, the 18 and 24-months old mice were in early and late aging respectively. Did the authors consider to utilize other structural/physiological indicators to determine the aging state for each animal?

Age ranges for the “early” aging group at 18 months and “late” aging group at 24 months were determined in accordance with the Jax “Lifespan as a biomarker criteria” (Flurkey et al.), which includes 18-24 months as the “old” stage within the mouse lifespan. 24 months of age was determined to be late aging without a significant death rate, as mouse survivorship steadily declines between 24 to 31 months of age (Flurkey et al.). There are also discrepancies in the literature regarding age stage designation for 18 months of age as being “aged” or “middle aged” when related to human age (PMID: 32641073), as such we defined this group as “early” aging. We use chronological age as our primary determinant of aging without considering potential individual variability based on other structural/physiological indicators. Rather, the goal of our study is to find robust differences in cerebrovasculature-related changes per age group. However, we did exclude animals with significant health concerns, particularly in our aged groups.

We added the following text in the introduction of the manuscript.

“...at 18 months (early aging), and 24-month-old (late aging) following the JAX Lifespan as biomarker criteria³⁰. 24 months of age was deemed as late aging without a significant death rate, as mouse survivorship steadily declines after 24 months old³⁰.”

2) The analysis of the changes in vascular density and capillary pericytes primarily utilized 18-month-old mice, while the examination of other vascular components and mural cells primarily utilized 24-month-old mice. What is the contribution of the vessel structural alterations to the changes in physiological parameters during late aging? The vessel structural data of 24-month-old mice needs to be provided.

In the late aging data, we found significant tortuosity in the penetrating arterioles. In the “Cerebrovascular structural changes with selective pericyte reduction in aged brains” section of the discussion, we provided the following text to link the vessel structural alterations with changes in physiological parameters in the discussion.

“.....aged brains showed substantially more tortuous penetrating arterioles, which impede blood flow by increasing flow resistance. This increase in resistance, unless countered by an increase in blood pressure, could result in reduced oxygen and nutrient supply, particularly in distal areas from main arteries such as the deep cortical layers and white matter tracks^{20,37,83}. Importantly, both human and murine studies have shown similar changes with tortuous vasculature, decreased vascular density, and associated slowed cerebral blood flow^{20,22,81,84,85}. Such changes can lead to an increased heart rate to compensate for cerebral hypoperfusion, as frequently observed in elderly population⁸⁶.”

We also performed the vascular structure data analysis with 24-month-old mice in comparison to 2-month-old mice using samples from tissue clearing and LSFM imaging. We found similar patterns as we observed in our STPT based imaging analysis. For instance, we found overall no significant vascular density changes in the cortical areas and selective reduction of vascular length density only in the cortical layer 6. Notably, we found a significant reduction in the infralimbic cortex in the 24-month-old mice, suggesting that this key area in the medial prefrontal cortex might be more vulnerable in the late aging. We added the following text in the result with the supplementary Figure 3.

“Lastly, we performed vascular tracing using pan-vascular lectin labeling and found no significant difference of vascular length density in the isocortex except a significant reduction in the infralimbic cortex in the 24-month-old mice (Supplementary Figure 3). Moreover, we found that only the cortical layer 6 shows a significant reduction in the vascular length density in the 24-month-old mice (Supplementary Figure 3), similar to changes identified in the 18-month-old mice (Figure 2N).”

3) The vascular parameters and components, such as artery radius and vascular tortuosity, were quantified using 3D immunostaining and tissue clearing in late aging mice. The tissue deformation rate, especially the vascular deformation rate for different aged samples, should be taken into consideration.

We acknowledge that the process of iDISCO tissue clearing does cause some tissue shrinkage globally, by approximately 30%, this is relatively similar across ages. While this shrinkage is unavoidable, we compare relative changes between samples from young and old brains with the same tissue clearing and the relative differences between the two age groups remain largely valid.

To address the issue of vascular deformation, we compared overall vessel shape and diameter in the same animal between *in vivo* 2-photon imaging and *ex vivo* iDISCO processed/LSFM imaging for a 24 month old sample. We confirmed that overall geometry of the vascular remains intact after the tissue clearing while the vascular diameter shrinks by 36%, which is largely similar to the overall brain volume change.

This result is now included as supplementary figure 2. We added the following text in the result section and added more relevant text in the method section.

“Despite the volume shrinkage due to dehydration-based tissue clearing methods, we confirmed that the overall vascular geometry was maintained by comparing *in vivo* two-photon and LSFM imaging from the same animal (Supplementary Figure 2).”

4) An interesting point that needs to be discussed, Fig. 2C shows that the volume of the isocortex shrinks in aged mice, which is a vulnerable region in aging. However, the degeneration of vessels occurred only in layer 6 (Fig. 2K-N), especially layer 6b nearby white matter. The pericyte density showed decrease in layer 6b too (Fig. 3), while the layers 2/3 and 4 showed significant increases. But the *in vivo* imaging only recorded superficial vascular signals. Which structural change has a greater impact? Or what is the impact of changes in layers 2 and 3 on the blood pressure and other parameters?

The isocortex volume is slightly increased (rather than decreased) from young (113.43mm³) to old (114.51mm³), but this is not significant. We updated Fig 2C to illustrate individual volume changes more clearly.

The reductions in vascular and pericyte densities within the deep cortical layer, particularly layer 6b likely affect the blood flow in both deep and superficial layers of the aged mouse cortex. Indeed, a recent study utilizing 3-photon *in vivo* imaging demonstrated reduced vascular density with vascular network simplification and increased flow resistance in layer 6 and the white matter track, while there are increases in baseline RBC flow velocity and flux in the superficial layer (<https://www.biorxiv.org/content/10.1101/2024.02.11.579849v1>). In parallel, our *in vivo* measurement for evoked neurovascular coupling (Figs. 6 and 7) showed that functional hyperemia in the superficial layer is largely intact, while blood oxygenation is significantly reduced. Collectively, this evidence suggests that the detrimental effects of aging occur most significantly in the deep cortical layer and may trigger potential compensatory response (e.g., increased density of pericytes) in superficial layer, which can lead to a redistribution of blood flow toward the

superficial cortical layer (<https://www.biorxiv.org/content/10.1101/2024.02.11.579849v1>). Coupled decreased oxygen delivery, these changes upon aging will make deep cortical layers highly vulnerable. We added the following text in the discussion.

“We also found significant reductions in vascular and pericyte densities within the deepest cortical layers, particularly layer 6b that are likely related to the blood flow responses in both deep and superficial layers of the aged mouse cortex. A recent study utilizing 3-photon *in vivo* imaging demonstrated reduced vascular density with vascular network simplification and increased flow resistance in layer 6 and the corpus callosum of the somatosensory cortex, while there are increases in baseline RBC flow velocity and flux in the superficial layer³⁷. In parallel, our *in vivo* measurement for evoked neurovascular coupling (Fig. 6 and 7) showed that functional hyperemia in the superficial layer is largely intact while RBC oxygen-carrying capacity is significantly reduced. Collectively, this evidence suggests that the detrimental effects of aging occur most significantly in the deep cortical layer and may trigger potential compensatory response (e.g., increased density of pericytes) in the superficial layer, which can lead to a redistribution of blood flow toward the superficial cortical layer³⁷. Coupled with less efficient on-demand oxygen delivery, these age-related changes will make deep cortical layers highly vulnerable. There is evidence that exercise can help ameliorate age-related declines in perfusion and oxygenation¹⁰⁷.”

5) It's more interesting to couple the physiological changes and vessel structural changes in the same mouse, for further modeling the effect of vascular structural changes on vascular functions. In line 325 and 327-329, the authors mentioned that “the aging brain has lower baseline oxygenation” “We observed that the oxygen delivered to the brain is significantly smaller in the aged mouse brain”. However, in Fig. 7F, the ΔHbT parameter in somatosensory cortex of 24-month-old mice was higher than that of 18-month-old mice. How many animals were used in these experiments (Fig. 6 and Fig. 7)? Were they also be analyzed in Fig. 7 C and D? Moreover, did the authors use the same animals in Fig. 6 and Fig. 7?

To extract information regarding the brain blood volume and brain tissue oxygenation responses, we imaged the awake mouse brain during different experimental procedures using spectroscopy. For this, ΔHbT (a ratiometric parameter) provides information about the relative change of brain blood volume in response to different experimental manipulations (e.g., voluntary locomotion, breathing pure oxygen). Another measurement is $\Delta\text{HbO}-\text{HbR}$ which provides information regarding the relative change of oxygenation in response to experimental manipulations.

In the original figure, when mice breath in 100% oxygen, ΔHbT in somatosensory cortex is relatively greater in 24 months old compared to 18 months old but this difference did not reach statistical significance. In the current figure, 24-month-old data has been removed. Nevertheless, the main message remains the same that old mice (18-month-old) struggle to deliver enough oxygen to the brain.

The same animals were used for both Fig. 6 and 7. 23 mice were used (2-4 months: 7 mice; 18 months: 5 mice) for Fig. 6 and 13 mice in total (2-4 months: 4 mice; 18 months: 4 mice) were used for Fig. 7. This information has been included in the methods section as well as results section.

6) In terms of age group, aged mice were divided into early old age (18-month-old) and late aging (24-month-old). However, in whisker stimulation experiment, the data of 18-month-old is lacking (Fig 6).

We measured a small set of animals ($n = 3$) with age around 18-months-old in the whisker stimulation experiment. We found no significant difference in 18-months-old compared to 2-months-old group. We removed the 24-months-old group.

7) In Fig. 1 “3-pericyte cell counting”, the signal detection of FITC with binarization appear to be positional bias.

We have removed the artifact in the pericyte counting panel.

8) For “in vivo imaging”, forelimb/hindlimb part of somatosensory cortex and anterior lateral motor cortex are chosen be the targeted area. There were differences between the results of two areas. Did the location of the observation window in the experiment affect the results? The authors should specify the relevant parameters in the methods.

The brain hemodynamics response to locomotion is brain region dependent, which is observed in previous studies from our group (Huo et al., *Journal of Neuroscience*, 2015, PMID: 25467301; Zhang et al., *Nature Communications*, 2019, PMID: 31797933) and other groups (Bergel et al., *Nature Communications*, 2020, PMID: 33273463). This brain region-dependent hemodynamic response to stimulation is also observed in fMRI studies. In our study, we did not see a significant difference in hemodynamic responses between 18-months-old and 2-month-old mice in both brain

regions. However, we found a significant reduction of oxygen-carrying capacity in 18-month-old brain only in the somatosensory but not in the frontal cortex, suggesting some regional differences.

For this study, we focused on two different brain areas, the somatosensory cortex (especially the forelimb/hindlimb representation of the somatosensory cortex, FL/HL) and anterior lateral motor cortex (ALM), as they are involved in motor planning and execution. When we analyzed our data, we drew regions of interest (ROI) based on standard stereotaxic coordinates and histological reconstruction (based on cytochrome C oxidase staining). The coordinates (relative to the bregma) of the region we targeted are as follows: for FL/HL (0-1 mm caudal and 1-2 mm lateral from bregma, ~1 mm²) and for frontal cortex (or ALM) (2-4mm rostral and 0.5-2.5mm lateral from bregma, ~4mm²). We have updated the following text in the manuscript to clarify this.

In the method section,

"We focused on two different brain areas, the somatosensory cortex (the forelimb/hindlimb area, FL/HL) and anterior lateral motor cortex (ALM). The coordinates are as follows: 0-1 mm caudal and 1-2 mm lateral from bregma, ~1 mm² for FL/HL and 2-4mm rostral and 0.5-2.5mm lateral from bregma, ~4mm² for the ALM."

9) In Fig 5I LSFM images, immunostaining lacks cell nucleus localization information of mural cell.

We used the combination of widely accepted markers for mural cells (CD13 and PDGFRb) that are cytoplasmic rather than nuclear markers. We have adjusted the arrow in Figure 5L to ensure that it is pointing to the cell body and increased the contrast of the figure to improve visibility.

Reviewer #2 (Remarks to the Author):

Review:

Aging drives cerebrovascular network remodeling and functional changes in the mouse brain

Authors:

Hannah C. Bennett, Qingguang Zhang, Yuan-ting Wu, Uree Chon, Hyun-Jae Pi, Patrick J. Drew, Yongsoo Kim

Summary:

The authors use mesoscale microscopy to determine vascular length and pericyte density changes at different brain regions in aged mice. They further show changes in hemodynamic responses to sensorimotor stimulations using two photon-imaging in awake aged mice.

Comments:

1. Authors are encouraged to expand on the novelty aspects of the manuscript. For example, it would be great to see a discussion how this work advances or diverges from previous studies that performed 3D brain vascular imaging in adult and aged mice (e.g., PMID: 30007164, PMID: 23093067, PMID: 32641073, PMID: 35022482)

We appreciate the suggestion of these excellent papers that will help to further support our work. Discussions regarding how this work contributes to and advances the existing body of work has been integrated into the discussion with the following text:

"This is particularly striking as the thalamus has recently been identified as a region vulnerable to microbleeds in aged mice ⁹⁸"

". In addition, aged mice have impaired hypoxia-induced vessel formation, which is a known compensatory mechanism ¹⁰⁵."

"Indeed, recent studies have demonstrated *in vivo* that aged animals, beginning at 18 months of age and becoming prominent at 24 months, have decreases in CBF correlated with areas of increased tortuosity in cortical areas ^{20,82}"

2. Vascular readouts: Additionally, to the vascular length and branching density it may be interesting to look at the nearest neighbor distance (NND) and distribution of between the individual vessels. These parameters have been shown to be important determinising regional hypoxic areas (PMID: 26567707, PMID: 32265643)

We performed the NND analysis and found that only the layer 6 showed significant increase in the isocortex. We created Supplementary Figure 1 to explain the result and added the following text in the result section.

“Moreover, we quantified nearest neighbor distance to vessels as a metric to access blood supply and found its significance increase in layer 6 in 18-month-old mouse brain (Supplementary Figure 1).”

3. Two-month old mice seem more of an adolescent age than fully adult stage of the mice. Usually, mice from 3 months are considered adult (e.g., according to JAX website). Also, some of the aged mice were 18 months that were used in some experiments; this may represent an age equivalent to a 55 year old human; that should be further discussed.

We acknowledge that 2 month old mice are younger than the age typically considered to be the mature adult. Hence, we elected to designate this as “young adult” to more accurately reflect their age.

Age ranges for the “early” aging group at 18 months and “late” aging group at 24 months were determined in accordance with the Jax Lifespan as a biomarker criteria (Flurkey et al.), which includes 18-24 months as the “old” stage within the mouse lifespan. 24 months of age was determined to be late aging without a significant death rate, as mouse survivorship steadily declines between 24 to 31 months of age [cite above]. While this criteria includes both of these groups in the “old” life stage, previous literature indicates that 18 and 24 months reflect differences in terms of brain aging (PMID: 23093067).

There are also discrepancies in the literature regarding age stage designation for 18 months of age as being “aged” or “middle aged” when related to human age (PMID: 32641073), as such we defined this group as “early” aging. We use chronological age as our primary determinant of aging without considering potential individual variability based on other structural/physiological indicators. Rather, the goal of our study is to find robust differences in cerebrovasculature-related changes per age group. However, we did exclude animals with significant health concerns, particularly in our aged groups.

We added the following text in the introduction of the manuscript.

“...at 18 months (early aging), and 24-month-old (late aging) following the JAX Lifespan as biomarker criteria ³⁰. 24 months of age was deemed as late aging without a significant death rate, as mouse survivorship steadily declines after 24 months old ³⁰.”

4. Both sexes of animals were used but analysed only in a subset of the experiments (Fig. 5). However, it is known that differences in vascular anatomy and CBF exist in mice and human (e.g., PMID: 27629099; PMID: 27844273). Did authors observe any region specific changes in vascular length/branching in the analysis Fig. 2-3?

Although sex differences may exist in cerebrovascular anatomy, the expected effect size is quite small. Indeed, we did not identify differences between the means of male and female vascular density/branching density/ average radii or pericyte density in either the young adult or aging groups. Ensuring the statistical power necessary to discern the small differences between sexes would require exceedingly large sample sizes (more than double of our current sample numbers), which is not feasible in this study. However, to facilitate future studies to resolve such potential differences, we have updated all figures, where feasible, to show the sex of each individual in each age group.

5. Did authors observe any leakage of the vascular tracer in aged mice? (PMID: 32641073). In my view, it is important to understand if the described selective reduction of pericyte density (reported e.g., in Fig. 3 and Fig. 5) leads to functional effects such as increased BBB leakage. Did the authors look at presence of tight junction proteins in these areas?

Thank you for raising this important point given the role of pericytes in BBB maintenance. We performed additional experiments to further interrogate this question regarding the age-related differences in tight junction proteins in regions with reduced pericyte density. We observed immunoglobulin extravasation not only in the deep cortical layer but throughout the entire layers of SS. This suggests that the pericyte function to maintain barrier properties is compromised upon aging even without loss of their regional density. We did not find any significant difference in the expression of ZO-1 protein, a tight junction marker in aged brains, which agrees with previous papers (PMID: 25784952; PMID: 30196051). We added the following text in the result section and updated our Figure 5 and add supplementary figure 5.

“Since pericytes play a key role in regulating blood-brain barrier (BBB) properties ^{16,46}, we examined whether aged brains show leaky BBB properties with compromised tight junction proteins. We analyzed serum protein extravasation in the somatosensory cortex (SS) ⁵⁷and found a significant increase in immunoglobulin extravasation throughout the entire layers of the SS (Figure 5M-N), in agreement with an in vivo study ⁵⁸. However, we did not find significant

changes in the expression of ZO-1 protein, a tight junction marker, in the SS of the aged brain (Supplementary Figure 5). This suggests that the pericyte function to maintain barrier properties is compromised upon aging even without loss of their regional density."

6. Several studies have previously shown that reduced pericyte coverage leads to BBB leaks. Authors should consider to cite and discussed these studies (e.g., PMID: 21040844, PMID: 31235908).

We added suggested papers in the result and discussion section where we provide evidence of leaky BBB in relationship with pericytes in aged brains, including.

Result section;

"Since pericytes play a key role in regulating blood-brain barrier (BBB) properties^{16,46}, we examined whether aged brains show leaky BBB properties with compromised tight junction proteins. We analyzed serum protein extravasation in the somatosensory cortex (SS)⁵⁷ and found a significant increase in immunoglobulin extravasation throughout the entire layers of the SS (Figure 5M-N), in agreement with an *in vivo* study⁵⁸."

Discussion section;

"Notably, pericytes are known to regulate the basal tone and permeability of microvessels, and release a neurotropic factor^{13,15,16,46}, and a recent study showed that pericytes in superficial cortical layers have impaired recovery of cellular processes in the aged brain³⁶. We found compromised BBB with increased immunoglobulin extravasation throughout cortical layers in aged brains⁵⁸. Therefore, while pericyte cell density does not change significantly during aging, their regulatory function may be impaired, resulting in slightly dilated and leaky cerebrovasculature."

7. Authors claim an increase of total brain volume by 6%, is this result statistically significant? How many animals were used and what was the standard deviation? The visualization in Fig 2B,C does not provide the information. Same goes for Fig 2D-F; is the decrease 10-20% statistically significant? In general, more information about the statistical analysis in Fig. 2,3 would be helpful. Did the authors correct their statistical method for each tested brain region?

Volume increase upon aging was not significant. All of the animals used for vessel analysis (n=4 young adult and n=5 aged) were included for determination of brain volume. We have incorporated this information in the figure depiction. The standard deviation for brain volume has been included as a value above each proportioned region in Figure 2B,C. Areas with statistically significant difference were highlighted with magenta boxes. Moreover, we provide Supplementary Data 1 (for vasculature) and Supplementary Data 2 (for pericytes) which contain mean, standard deviation, un-corrected p-value, and multiple comparison corrected q value based on False Discovery Rate (FDR). Finally, statistical section has also been updated to include more details. All statistical analysis was corrected for multiple comparisons in Prism using the Two-stage step-up method of Benjamini, Krieger and Yekutieli in Graphpad to account for comparing across multiple brain regions.

8. It would be interesting to know if the aging-induced remodeling in arterioles are associated with differences in functional readouts such as CBF (Fig.5). Did authors perform any CBF readouts of whole mouse brains? Such as Laser Speckle/Doppler Imaging/fMRI?

Being able to quantify blood flow response across the whole brain would be ideal. However, the suggested methods, such as laser speckle/doppler imaging/fMRI can only provide a ratiometric quantification, not absolute, quantitative flow measurements, which may only provide partial information about the functional change during aging. While we did not specifically perform CBF readouts of the whole mouse brains, several studies have previously shown *in vivo* that 18 and 24 month old mice do have decreases in CBF in cortical areas shown to have increased tortuosity (PMID: 35022482, PMID: 30007164:). In our study, we did measure red blood cell velocity in capillaries and small arterioles, which provides an absolute value of how fast the blood flow through the vessels. We found a trend toward decreased RBC velocity, but not a statistically significant difference (2-month: 0.58 ± 0.33 mm/s; 18-month: 0.53 ± 0.34 mm/s) (Supplementary Figure 6).

We added the following comments in the discussion, reflecting prior studies.

"The lateral ENT (ENT1) in aged brains showed significantly decreased vascular length, branching point, and capillary pericyte density, consistent with *in vivo* ultrasound measurement⁸²."

"Indeed, recent studies have demonstrated *in vivo* that aged animals, beginning at 18 months of age and becoming prominent at 24 months, have decreases in CBF correlated with areas of increased tortuosity in cortical areas ^{20,82}."

Minor

1. The plots in Fig 5 C,G, J, N are too small

The sizes of these plots have been increased.

2. Statistical section should be explained in more detail and significance should be indicated consistently in all Fig panels.

We added more details in the statistical section and updated relevant figures (Figure 2,3,5) and added supplementary Data 1 and 2 with more detailed information.

Reviewer #3 (Remarks to the Author):

This is a comprehensive assessment of brain-wide cerebrovascular structural and functional changes with normal aging in mice. The authors assess multiple parameters of cerebrovascular structure at three ages (2, 18 and 24 months of age) and do so in both sexes on mice. They do a fair (in most cases) assessment of their results and compare these findings to those published for individual brain regions. In brief, this is a comprehensive assessment that has never before been reported.

A few weaknesses are noted below:

1. Both sexes are included in all studies, but on only one occasion are sexes separated in presentation of the data and/or discussed in interpretation of the results. In the one case where sex of mice was shown, there are too few animals in each group to test for sex differences.

Although sex differences may exist in cerebrovascular anatomy, the expected effect size is quite small. Indeed, we did not identify differences between the means of male and female vascular density/branching density/ average radii or pericyte density in either the young adult or aging groups. Ensuring the statistical power necessary to discern the small differences between sexes would require exceedingly large sample sizes (more than double of our current sample numbers), which is not feasible in this study. However, to facilitate future studies to resolve such potential differences, we have updated all figures, where feasible, to show the sex of each individual in each age group.

2. The authors use the terms "early aging" (18 months) and "late aging" (24 month), without a definition of these terms. How physiologically or pathophysiologically are mice at these ages different? A statement about where these two ages are relative to the median life span of the mice strain would be appropriate.

We added the following text in the introduction of the manuscript.

"...at 18 months (early aging), and 24-month-old (late aging) following the JAX Lifespan as biomarker criteria ³⁰. 24 months of age was deemed as late aging without a significant death rate, as mouse survivorship steadily declines after 24 months old ³⁰."

3. In studies where a subset of mice was selected for assessments from a larger group under investigation, how was this selection made? This is important as it needs to be done randomly and irrespective of the data from the larger group. This selection process should be stated in the manuscript.

We initially acquired LSFM data with vascular labeling (n = 9 each for 2months and 24 months old). However, we used a subset of samples in the downstream analysis based on immuno staining qualities of target signals. For samples passing quality control, we randomly selected a small number of samples due to the required intense labor of manual analysis.

We added the following text in the method section.

"We initially acquired LSM data with vascular labeling (n = 9 each for 2 months and 24 months old) and used them for Circle of Willis analysis. we used a subset of samples randomly chosen from data with satisfactory immunostaining qualities of target structures."

4. The authors state "Adult 2-month-old C57BL/6J mice were bred from C57BL/6J mice directly obtained from the Jackson Laboratory and used for vascular tracing experiments with FITC filling (n=4). 18-month-old C57BL/6J mice utilized for FITC-fill vascular mapping experiments were aged from a local C57BL/6 mouse colony. 24-month-old C57BL/6J mice used for the current study were directly obtained from the National Institute of Aging at 18 months and aged to 24 months in house." As such, it appears that each age group may have had a difference housing experience (housing environment like lighting and food source), shipping and handling. Could these factor contribute to the apparent aging effects reported? The authors failed to mention this in the potential limitation of the study.

Thank you for this comment as it helped to identify a mistake within the methods section as 18-month-old C57BL/6J mice utilized for FITC-fill vascular mapping experiments were obtained directly from Charles River. However, we do acknowledge that 24month old mice initially obtained at 18months and aged to 24months in house does confer potential limitations and confounders regarding animal stress and nutrition. Therefore, we have incorporated this as a potential limitation of the work in the discussion section.

"Another potential limitation to this work is that most C57BI/6 aged animal groups were obtained directly from commercial vendors which could confer differences in housing, nutrition, shipping, and handling compared to the young adult mice raised in-house. While this is unlikely to overshadow the differences due to aging, these factors could potentially impact the vasculature."

REVIEWER COMMENTS

Reviewer #1 (Remarks to the Author):

I have seen the efforts of the authors to revise the manuscript and supplements, which make the study more concrete and rigorous. I have a few suggestions.

1) To group the mice in different ageing period, the authors should consider the individual differences in physiological states and gender, which may affect the overall level. As shown in Supplementary Figure 5C, there is an extreme sample in the 24M group, which may affect the groups. The authors can make the assessment by adding other parameters, referring to indicators such as behavior and metabolism to divide the age into more standard (e.g., PMID: 37857723; PMID: 22355651; PMID: 22953032).

2) Based on the authors' statistical analysis, it appears that capillary pericytes in some brain regions have changed significantly in early aging, while mural cells only begin to change in late aging. Is there a sequential order of aging of the different vascular components?

3) As the authors mentioned " overall geometry of the vascular remains intact after the tissue clearing while the vascular diameter shrinks by 36%", does the loss of vascular signals cause false-positive results of vascular connectivity damage? The authors should certify with continuous inspections in their 3D datasets.

4) The author deleted all the in vivo experimental data of 24-month-old mice, retaining vascular function monitoring in the early aging stages (18M). Whether there are gender-dependent functional differences in aged mice? The authors should list the sex of all mice especially in Fig 6-7.

5) The authors found many vulnerable areas, such as the medial prefrontal cortex, somatosensory cortex, and the deep layer near the corpus callosum. The author could sort out the rules with mathematical construction method (PMID: 37745386) to sort out the rules, such as whether the vascular remodeling caused by aging is connection or function related.

Reviewer #2 (Remarks to the Author):

I thank the authors for addressing my comments and making the necessary revisions to the manuscript. All points have been satisfactorily resolved.

Reviewer #3 (Remarks to the Author):

The authors have responded successfully to each of my four main issues. Inasmuch as this study is a detailed anatomical assessment, the use of sufficient number of male and female subject is not likely, although both sexes were incorporated into the design of the study. I suggest that a statement to this effect be added to the manuscript.

REVIEWER COMMENTS

Reviewer #1 (Remarks to the Author):

I have seen the efforts of the authors to revise the manuscript and supplements, which make the study more concrete and rigorous. I have a few suggestions.

We appreciate the positive evaluation from the reviewer.

1) To group the mice in different ageing period, the authors should consider the individual differences in physiological states and gender, which may affect the overall level. As shown in Supplementary Figure 5C, there is an extreme sample in the 24M group, which may affect the groups. The authors can make the assessment by adding other parameters, referring to indicators such as behavior and metabolism to divide the age into more standard (e.g., PMID: 37857723; PMID: 22355651; PMID: 22953032).

We agree that aging may affect individual animals differently as suggested by papers referenced by the reviewers. To link such individual variabilities and divide aged animals into different subgroups based on behavioral and other physiological parameters will require a completely new set of experiments with a larger sample size, which is not feasible in the current scope of the manuscript. However, we added the following sentence in the "Limitations of the study" of the discussion section.

"Additionally, aging could affect individual animals and animals with different sexes differently. Hence, future studies that include additional behavioral and physiological measurements from both sexes with sufficient animal numbers will provide further insight to link vascular changes with individual variabilities and sex differences upon aging."

2) Based on the authors' statistical analysis, it appears that capillary pericytes in some brain regions have changed significantly in early aging, while mural cells only begin to change in late aging. Is there a sequential order of aging of the different vascular components? Mural cell types include vascular smooth muscle cells and pericytes (and their various morphological subtypes). Moreover, capillary pericytes are the most predominant pericytes in the brain. Our primary measurement for pericytes focuses on the density of capillary pericytes, which are impacted by both early and advanced aging.

In Figure 3, we presented evidence that 18-month-old brain showed a selected reduction of capillary pericyte density in the basal forebrain and layer 6 of the isocortex compared to 2-month-old brain using PDGFR β -Cre;Ai14 reporter mice and STPT imaging. Separately, we used targeted region of interest-based quantification using C57 mice with tissue clearing, 3D immunolabeling, and LSFM imaging in Figure 5H-L and Supplementary Figure 4. We found a selected reduction of layer 6 pericyte density in the somatosensory cortex of 24-month-old mice (Figure 5J-L) while the same area did not show significant reduction in the 18-month-old data (Figure 3J). Similarly, the entorhinal cortex (ENT) showed a significant reduction of capillary pericytes in 24-month-old mice (Supplementary Figure 4) while the same area did not show a significant reduction in the 18-month-old data (Figure 3B). In contrast, ensheathed pericytes that reside between arteries and microvessels did not show significant reduction even in 24-month-old mice (Supplementary Figure 4). Collectively, these data showed that capillary pericytes are vulnerable upon aging and advanced aging (24-month-old mice) showed further reduction of capillary pericyte density compared to early aging (18-month-old mice).

The summary of this result was included in the manuscript.

"This suggests that capillary pericytes are at higher risk of cellular density loss, particularly in advanced age."

3) As the authors mentioned "overall geometry of the vascular remains intact after the tissue clearing while the vascular diameter shrinks by 36%", does the loss of vascular signals cause false-positive results of vascular connectivity damage? The authors should certify with continuous inspections in their 3D datasets.

Indeed, our vascular length measurement occur with continuous inspections in their 3D datasets. The volume shrinkage due to the iDisco-based tissue clearing did not affect vascular labeling. Moreover, we use Elastix-based registration tool to precisely quantify volume of individual brain areas in order to calculate densities. We added the following text in the method section.

"We used TubeMap graph tracing tool to follow vasculature continuously to calculate vascular length in 3D volume. Moreover, we used Elastix-based image registration to calculate volumes of different brain regions and to map signals in the 3D reference brain."

4) The author deleted all the in vivo experimental data of 24-month-old mice, retaining vascular function monitoring in the early aging stages (18M). Whether there are gender-dependent functional differences in aged mice? The authors should list the sex of all mice especially in Fig 6-7.

Although sex differences may exist, the expected effect size is quite small. Indeed, we did not identify differences between the means of male and female vascular anatomy and functional data. Ensuring the statistical power necessary to discern the small differences between sexes would require exceedingly large sample sizes (more than double of our current sample numbers), which is not feasible in this study. However, to facilitate future studies to resolve such potential differences, we have updated figures including Fig 6-7 to show the sex of individual animals in data plots wherever feasible, and specified animal sex in the result, method, and figure legend.

5) The authors found many vulnerable areas, such as the medial prefrontal cortex, somatosensory cortex, and the deep layer near the corpus callosum. The author could sort out the rules with mathematical construction method (PMID: 37745386) to sort out the rules, such as whether the vascular remodeling caused by aging is connection or function related.

The main objective of the current manuscript is to generate data resources to comprehensively examine altered vascular network in the

whole brain and to quantify functional changes in the selected cortical area. Using these data, we postulated how these changes can impact specific brain functions in connection with various pathological conditions in the Discussion. For instance, we have the following paragraph to link how areas identified in the current studies can help to explain the regional vulnerabilities in neurodegenerative disorders.

“Since our 3D mapping data examine vascular network changes of the whole mouse brain in an unbiased way, we identified specific brain regions with selective vulnerabilities in aged brains. For example, we found significantly reduced vascular and pericyte densities in the basal forebrain area, which contains cholinergic neurons⁹¹. The basal forebrain cholinergic neurons (BFCNs) have highly extensive projections to the cortical area and have large soma size with high energy demands⁹². Previous clinical and preclinical studies have shown that BFCNs are highly vulnerable in Alzheimer’s disease (AD) and their deterioration is linked with memory impairment^{44,93}. Impaired vascular networks with decreased pericyte density may, potentially serve as an underlying cause of BFCN degeneration in normal aging and neurodegenerative disorders, including AD^{79,94,95}. Another notable area is the entorhinal cortex (ENT), a part of the hippocampal network, which has been heavily implicated in AD and particular cognitive deficits^{96,97}. The lateral ENT (ENTl) in aged brains showed significantly decreased vascular length, branching point, and capillary pericyte density, consistent with *in vivo* ultrasound measurement⁸². The ENTl vascular density is one of lowest across the brain region in normal adult mice²⁹. This indicates that the entorhinal cortex is likely to be particularly susceptible to further insult (e.g., hypoxia), which may explain its vulnerability in neurodegenerative disorders. Lastly, our study identified specific thalamic and hypothalamic areas with decreases in the vascular network density, such as the medial preoptic area. This is particularly striking as the thalamus has recently been identified as a region vulnerable to microbleeds in aged mice⁹⁸. This warrant future studies for these largely understudied subcortical areas in aging research.”

Although building a mathematical model to explain how vascular structural changes can impact specific functions will be highly informative, we believe that such work is beyond the scope of the current manuscript. However, acknowledging the importance of such future work, we added the following sentence in the discussion.

“Moreover, another potentially informative future direction is to utilize mathematical models to delineate how vascular remodeling in aging is linked with vascular connectivity or functional impairment.”

“These changes can explain the vulnerability and resilience of different brain areas in normal aging, which will inform future experimental and computational approaches to gain a comprehensive understanding of brain aging.”

Reviewer #2 (Remarks to the Author):

I thank the authors for addressing my comments and making the necessary revisions to the manuscript. All points have been satisfactorily resolved.

We appreciate the positive evaluation from the reviewer.

Reviewer #3 (Remarks to the Author):

The authors have responded successfully to each of my four main issues. Inasmuch as this study is a detailed anatomical assessment, the use of sufficient number of male and female subject is not likely, although both sexes were incorporated into the design of the study. I suggest that a statement to this effect be added to the manuscript.

We appreciate the positive evaluation from the reviewer. We added the following sentence in the discussion to highlight limitation of the current study.

“Additionally, aging could affect individual animals and animals with different sexes differently. Hence, future studies that include additional behavioral and physiological measurements from both sexes with sufficient animal numbers will provide further insight to link vascular changes with individual variabilities and sex differences upon aging.”

REVIEWERS' COMMENTS

Reviewer #1 (Remarks to the Author):

The authors have responded successfully to my main issues.